# Towards Perpetually Trainable Neural Networks

## Abstract

Underpinning the past decades of work on the design, initialization, and optimization of neural networks is a seemingly innocuous assumption: that the network is trained on a *stationary* data distribution. In settings where this assumption is violated, e.g. deep reinforcement learning, learning algorithms become unstable and brittle with respect to hyperparameters and even random seeds. One factor driving this instability is the loss of plasticity, meaning that updating the network's predictions in response to new information becomes more difficult as training progresses. In this paper, we conduct a thorough analysis of the sources of plasticity loss in neural networks trained on nonstationary learning problems, identify solutions to each of these pathologies, and integrate these solutions into a straightforward training protocol designed to maintain plasticity. We validate this approach in a variety of synthetic nonstationary learning tasks, and further demonstrate its effectiveness on two naturally arising nonstationarities: reinforcement learning in the arcade learning environment, and by an adaptation of the WiLDS distribution shift benchmark.

## 1 Introduction

Training a neural network on a single task is relatively straightforward: for popular data modalities it is usually possible to take an off-the-shelf architecture and optimization algorithm and, with minimal hyperparameter tuning, obtain a reasonably well-performing predictor. Yet the statistical relationships learned by machine learning systems are often dynamic: consumer preferences change over time, information on the internet goes out of date, and words enter and leave common usage. In reinforcement learning systems, the very act of improving the learner's policy introduces changes in the distribution of data it collects for further training. Without the ability to update its predictions in response to these changes, a learning system's performance will inevitably decline. These failures are often resolved in practice by training a new model from scratch on updated training data, a strategy that can be prohibitively expensive – consider the cost of re-training a foundation model to update its knowledge base – or impractical, as can be seen in the added complexity of distillation approaches used in some reinforcement learning (RL) agents.

The observation that training a neural network on some nonstationary prediction problems results in a reduced ability to adapt to new tasks has been made independently several times, both in reinforcement learning (Dohare et al., 2021; Igl et al., 2021; Kumar et al., 2020; Lyle et al., 2021; Nikishin et al., 2022; Abbas et al., 2023) and in supervised learning (Ash & Adams, 2020; Berariu et al., 2021). In spite of its prevalence, our understanding of why this phenomenon, which we will refer to in this paper as *plasticity loss*, occurs remains vague. Lyle et al. (2023) show a number of negative results indicating that we cannot attribute all instances of plasticity loss to a single straightforward statistic of the network parameters. Prior work has studied a number of diverse strategies for maintaining plasticity: for example, Sokar et al. (2023) reset dormant neurons on a subset of Atari games, whereas Ash & Adams (2020) shrink and perturb parameters when data is added to an image classification problem. Each of these strategies is designed to address a particular mechanism of plasticity loss that is largely distinct and non-overlapping with other possible causes: resetting dormant neurons will avoid plasticity loss due to saturated activation functions, but will do little to address an ill-conditioned loss landscape, whereas regressing a network's features towards their initial values will only have an effect if the network is effectively propagating gradients and will

## Mechanisms of plasticity loss

**Figure 1:** Summary of main pathologies identified that can lead to measured plasticity loss.

have no effect on dead ReLU units. A model of plasticity loss must take the independence of these mechanisms into account to produce effective mitigation strategies.

This paper aims to develop such a model, and use it to maintain plasticity indefinitely. Our contributions are as follows: first, we conduct a fine-grained empirical analysis into the process by which plasticity loss occurs in a variety of domains, and identify four largely independent mechanisms: unit saturation, preactivation distribution shift, unbounded parameter growth, and regression target scale (c.f. Figure 1) of which preactivation shift and regression target scale have not been previously identified as factors in plasticity loss. Our analysis of these mechanisms sheds new light onto the well-known phenomenon of dormant neurons(Sokar et al., 2023), revealing that saturated ReLU units are a symptom of a deeper underlying pathology in the network's optimization dynamics characterized by shifts in the distribution of preactivations, which can cause plasticity loss even in nonsaturating nonlinearities. We further show that the findings of Lyle et al. (2023), which suggested that plasticity loss in deep RL is due to loss landscape pathologies, can in fact be largely explained by the shift in the magnitude of regression targets in temporal difference algorithms. These findings reveal a more nuanced relationship between loss landscape curvature and plasticity than had been previously conjectured, where we find that curvature-aware optimizers can still encounter training difficulties in nonstationary tasks.

Critically, we show that mitigation strategies to these mechanisms can be combined to produce learners which are significantly more robust to nonstationarity than would be obtained by any single strategy in isolation. Whereas prior work indicated that each mechanism on its own cannot fully explain plasticity loss (Lyle et al., 2023), we find that addressing all mechanisms in conjunction is sufficient to yield negligible loss of plasticity in a variety of synthetic benchmarks, and that these methods can further improve performance on both deep reinforcement learning benchmarks and on natural distribution shifts. Our approach, which involves identifying effective interventions on each of the above mechanisms in isolation and then combining the best of each, significantly reduces the combinatorial complexity of the search over plasticity-preserving learning algorithms. The resulting framework unifies disparate findings from the broader literature on plasticity loss and illuminates promising directions for future work to identify even more effective mitigation strategies for the mechanisms we have identified.

## 2 BACKGROUND AND RELATED WORK

### 2.1 NETWORK TRAINING, SIGNAL PROPAGATION, AND PREACTIVATION DISTRIBUTIONS

The training dynamics of neural networks have been extensively studied over the past decades (e.g. Sutskever et al., 2013). This literature has highlighted a number of important design choices for neural network architectures, initialization schemes, and optimizers, which will come to bear on the analysis in this paper. Of particular relevance to this work is layer normalization (Ba et al., 2016), which normalizes vectors in a network (often preactivations) to have sample mean 0 and variance 1, commonly used activation functions like ReLU and tanh, and schemes designed achieve good signal propagation[1] via careful initialization (Poole et al., 2016; Balduzzi et al., 2017) or network design and transformation (Martens et al., 2021; Hayou et al., 2021; Zhang et al., 2021; He et al., 2023).

---

[1]Roughly speaking, networks with poor signal propagation don't do a good job of propagating geometric information about their inputs to deeper layers. This can lead to optimization difficulties, such as vanishing or exploding gradients, as has been formalized in Xiao et al. (2020) and Martens et al. (2021) in the "NTK regime".

In the signal propagation literature (e.g Daniely et al., 2016; Poole et al., 2016; Schoenholz et al., 2017; Lee et al., 2017; Xiao et al., 2018; Hanin & Nica, 2019; Hayou et al., 2019; Yang, 2019; Martens et al., 2021), the distribution of preactivations (i.e. the inputs to element-wise nonlinear layers) plays a crucial role in predicting the functional behavior, and ultimate trainability, of a randomly initialized network. In the most popular schemes to achieve good signal propagation, parameters are initialized as iid Gaussian with mean and variance chosen so that the empirical preactivation distribution within a layer resembles a Gaussian with some target mean $\mu$ and variance $\sigma^2$ (with high probability for all network inputs). The choices of $\mu$ and $\sigma^2$ are crucial, and their optimal values will depend on the characteristics of the activation function, as well as those of the network's architecture (Martens et al., 2021). While there can be no universally optimal choice[2] for $\mu$ and $\sigma^2$, the standard choice $\mu = 0$ and $\sigma^2 = 1$ will often be good enough for many commonly used activation functions, assuming modest depth or the use of residual connections (He et al., 2016). Activation functions like ReLU and tanh are arguably popular precisely because they work well with $\mu = 0$ and $\sigma^2 = 1$. And for ReLU in particular, signal propagation is essentially independent of $\sigma^2$ whenever $\mu = 0$ (Zhang et al., 2021), making ReLU networks more robust[3] to suboptimal choices of the initialization variance of the weights and biases (assuming a mean of zero). Notably, applying layer normalization to a preactivation vector ensures that the 1st and 2nd order sample statistics precisely agree with $\mu = 0$ and $\sigma^2 = 1$. While this won't guarantee the preservation of good signal propagation as training progresses, as this depends on various other conditions holding, it may indeed be helpful.

## 2.2 Loss of plasticity

A good initialization ensures that, at least when training begins, the network it is applied to is capable of learning – in other words, that it is *plastic*. As training progresses, however, it is possible for this plasticity to be lost (Dohare et al., 2021; Berariu et al., 2021; Abbas et al., 2023). The term plasticity has been used to refer to both a network's ability to attain low generalization error (Berariu et al., 2021), and its ability to improve its performance on a training set (Abbas et al., 2023; Nikishin et al., 2023; Kumar et al., 2023b). This paper will adopt the latter interpretation, focusing on mitigating the optimization difficulties observed in nonstationary learning problems. We will revisit several of the synthetic nonstationary classification tasks studied by Lyle et al. (2023) along with a more natural class of nonstationarities, where we will be interested in avoiding an increase in the final loss achieved immediately before a task change; this approach is slightly different from that of Kumar et al. (2023a), who look at average loss.

Several prior works have studied the mechanisms of plasticity loss and proposed novel mitigation strategies (Abbas et al., 2023; Nikishin et al., 2022; Dohare et al., 2021; Ash & Adams, 2020). The key insight provided by this work is that, to a large extent, several key mechanisms of plasticity loss are independent[4], and so plasticity loss can be addressed by a combination of mitigation strategies targeting each component mechanism. This observation explains the surprising results of Lyle et al. (2023), who argue that, taken in isolation, many network statistics cannot fully explain plasticity loss: plasticity can be lost via a variety of mechanisms, and mitigating one of these mechanisms does not entail mitigating the others. While some of the mechanisms we study such as dormant neurons (Sokar et al., 2023) have been noted by prior work (or indeed, in some instances 'disproved' as single explanations (Lyle et al., 2023)), we also identify new mechanisms such as the shift in the mean and norm of network features and the structure of the network's regression targets, and illustrate how these mechanisms can interact in unexpected ways.

## 3 Failure modes of nonstationary learning

We begin our analysis with a mechanistic study of learning dynamics in nonstationary image classification tasks. We will consider three primary sources of nonstationarity: permutation of the pixels

---

[2]To see this, note that if $\mu_0$ and $\sigma_0^2$ are optimal for a given activation $\phi_0$, then $\mu_1 = \mu_0 + 3$ and $\sigma_1^2 \equiv \sigma_0^2/10$ would be optimal for the activation defined by $\phi_1(x) \equiv \phi_0(\sqrt{10}\,(x-3))$.

[3]While signal propagation in ReLU networks is unaffected by the preactivation variance $\sigma^2$, activations can still explode or shrink as noted by He et al. (2015), leading to very large/small network outputs. Very large output can cause problems for many standard loss functions, such as softmax cross-entropy, unless one employs layer normalization or manual rescaling at the network's penultimate layer.

[4]For a more detailed discussion of what we mean by independence, c.f. Appendix E.

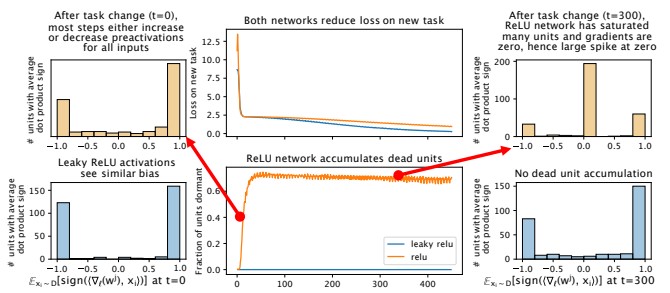 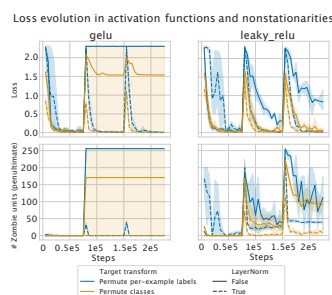

**Figure 2:** Accumulation of dead units after a task change: we train a small MLP on a random label memorization task, then visualize the learning curve when trained parameters are used as starting point for optimization on a new set of random labels. At the start of the new task, for many units in the first hidden layer their input weight gradients have negative dot product with *all* network inputs (c.f. insets of **left** side figure): if the network uses ReLU activations, this results in the accumulation of dead units (c.f. **right** side figure), and even non-saturating variants such as leaky-ReLU exhibit an analogous accumulation of 'zombie' units, where all inputs exhibit either to the right or left of the nonlinearity.

of the input images, permutation of the class labels, and uniform random relabeling of each example. Random re-labeling presents a particularly challenging form of nonstationarity. Because the inductive bias of a randomly initialized network is not typically aligned with the random labels, this task requires significant adaptation from the network each time the labels are re-randomized. In contrast, permuting class labels while keeping class membership fixed is an easier learning problem for which the network does not need to radically transform its representation. This section will identify four primary mechanisms by which networks lose plasticity: saturation of nonlinearities, distribution shifts in unit preactivations, uncontrolled norm growth, and loss landscape pathologies; c.f. Figure 1 for a visualization. Our objective is to understand how these mechanisms cause the loss of plasticity; while we do evaluate training protocols which can mitigate some mechanisms to facilitate this understanding, we defer a full evaluation of mitigation strategies to Section 4.

### 3.1 SATURATION OF NONLINEARITIES

Perhaps the most conspicuous form of plasticity loss we will study arises due to saturation of the non-linearities in the network (Lyle et al., 2023; Abbas et al., 2023), resulting in 'dead' (Lin et al., 2016) or 'dormant' (Sokar et al., 2023) units in the case of the widely-used ReLU and GeLU activations when all pre-activations are negative. A saturated activation function is unable to propagate gradients back to its incoming weights, which means that unless the unit's input distribution shifts in a suitable direction, the parameters associated with this unit will remain permanently frozen. Without intervention it is therefore possible for a network to gradually accumulate more and more dead units as it trains. In theory, one might expect this to occur simply as a result of a random walk in parameter space, where each unit has some probability of dying after each parameter update. In this section we will see that in fact particular forms of nonstationarity can bias the network towards updates which rapidly cause unit death.

To obtain a fine-grained view on unit saturation, we train a neural network to convergence on one set of random labels of CIFAR-10, and then re-shuffle the labels and continue optimization from the converged parameters and optimizer state. We observe in Figure 2 a marked increase in the number of dead units immediately after the task switch, followed by a re-equilibration period during which some units desaturate. We illustrate these two phases in the left-hand-side plot in figure 2, where we track the loss and number of dead units in a small MLP immediately after a task change in the random labels task. We see that initially the gradients on the incoming weights to the first layer tend to either increase or decrease the preactivation values of *all inputs to the unit*. This bias towards uniform effects of updates on the preactivation values in the immediate aftermath of a task switch can be explained by the pressure from the loss function to reduce the magnitude of incorrect logits and increase the entropy of the predicted distribution (c.f. Figure 16 in Appendix D for further details). Coupled with large gradient magnitudes, and hence large step sizes immediately after a task switch, this results in a high rate of units dying off as the optimizer takes large steps that set all preactivations below zero.

## 3.2 PLIGHT OF THE LIVING DEAD ReLU UNIT

In general, it is not desirable for a gradient step to have the exact same directional effect on a unit's output for all inputs. Most popular nonlinearities used in neural networks behave as a linear or constant function outside of a limited range, and recent initialization schemes explicitly make use of this property (Martens et al., 2021). If the preactivation distribution drifts outside of this range after initialization, for example due to training on nonstationary objectives, signal propagation in the neural network may degrade, inhibiting optimization. One extreme version of this arises in the case of dormant ReLU units, however analogous failures also exist: for example, a ReLU unit whose preactivation value is positive for all network inputs is effectively a linear function. Unlike dead units, through which gradients cannot flow by definition, these 'zombie units' can propagate gradients; however, because linear units are less expressive than nonlinear ones (e.g. Martens et al., 2013; Montufar et al., 2014; Raghu et al., 2017), the overall expressive power of the network is effectively reduced, at least transiently. While saturated units have received much attention as a factor in plasticity loss, the accumulation of effectively-linear units has not previously been studied in the context of network plasticity.

One simple solution to the dormant neurons seen in Section 3.1 is to use a non-saturating activation function, such as Leaky ReLU. However, as we saw in the leftmost plot of Figure 2, use of Leaky ReLUs does not prevent the optimizer from uniformly increasing or decreasing preactivation values (over all network inputs) at a task boundary. As a result, we see in the right hand side of Figure 2 that sudden task changes in the random label memorization task (distinguishable by the two sharp upticks in the loss that occur in each plot after the initial descent) induce a similar number of 'zombie units' for non-saturating activations as they do dormant neurons for saturating activations. Increased numbers of linearized units are consistently observed in parallel with reduced plasticity in both saturating and non-saturating activation functions; we provide additional examples of this trend in Appendix D.7, and illustrate a minimal example of a purely linear network which is unable to learn in Figure 22. The exact mechanisms by which these distribution shifts interfere with signal propagation in a network are complex and outside the scope of this work; we provide a brief discussion in Appendix 2.1. The accumulation of zombie units can therefore be interpreted as a signal that the network's training dynamics have gone awry, even if the causal relationship connecting zombie units and plasticity loss remains ambiguous. We can further conclude that while preactivation drift can result in dead units, it has other undesirable effects on signal propagation and trainability which are **independent** of its effect on dormancy.

## 3.3 NORM GROWTH

While prior works have observed that parameter norm growth often correlates with plasticity loss, the causality relating these phenomena has exhibited notable resistance to analysis. We observe two main learning difficulties associated with growth in the network parameters and outputs. The first is that as the parameter norm grows to extreme values, neural networks are vulnerable to numerical instability; this can be exacerbated by increased sharpness of the loss landscape which also frequently accompanies parameter norm growth. The second is that increasing parameter magnitude reduces the effective change in network predictions induced by a fixed-norm update.

We study the effect of norm growth in the iterated random label memorization tasks by training a variety of architectures on this domain for 15 million optimizer steps. We see in the leftmost plot of Figure 3 that increased parameter norm accompanies plasticity loss in all architectures we evaluate. Notably, there is not a monotone relationship between parameter norm at the end of training and plasticity: networks with all saturated units do not propagate gradients which could contribute to norm growth. In architectures incorporating normalization layers, it is not possible to saturate all units and so the parameter norm is able to increase by several orders of magnitude, in some cases entering the tens of millions by the end of training. This extreme growth in norm results in modest reductions in accuracy at the end of each task as training progresses. Mild L2 regularization (we use a value of 1e-5) is sufficient to resolve this issue in all three architectures. We also observe that increasing parameter norm corresponds to increasing sharpness of the loss landscape, where this is surprisingly most pronounced in the networks with layer normalization.

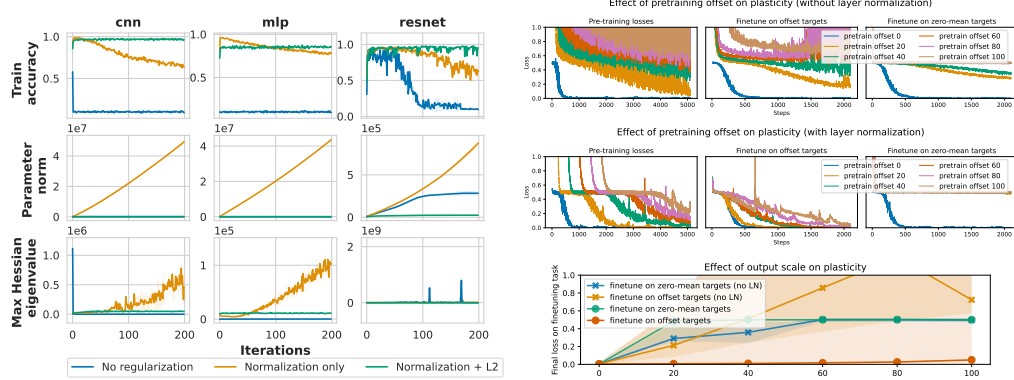

**Figure 3: Left.** Training networks on 200 task changes of the sequential label memorization task. Combining layer normalization and L2 regularization results in essentially no measurable plasticity loss even in the final iterations of our evaluation. **Right.** Illustration of the relationship between pretraining target magnitude and optimization speed on a new task. We see a strong dose-response effect from increasing the magnitude of regression targets on the final loss on a fine-tuning task, and observe similar trends in the learning curves on these tasks as those observed by Lyle et al. (2023) in DQN agents trained on contextual bandits.

## 3.4    OUTPUT SENSITIVITY

Our analysis so far has illustrated pathologies which are exacerbated by fairly generic forms of nonstationarity; however, most naturally-occurring learning problems evolve in a structured way. For example, in deep reinforcement learning, the temporal-difference (TD) targets towards which the value function is regressed tend to increase in magnitude over time. While some prior works have suggested that TD targets may induce overfitting in neural networks (Raileanu & Fergus, 2021), the issue of their increasing scale has not been considered as a possible impediment to optimization. van Hasselt et al. (2016) observed that normalizing regression targets can speed up convergence, but saw mixed effects of this normalization strategy in the Atari benchmark, which was attributed to the incidental exploration provided by the standard reward clipping strategy. In contrast, in this section we will argue that *regression targets with large non-zero mean are innately difficult for neural networks to fit in a way that preserves plasticity*.

A particularly simple example of this is the contextual bandit studied by Lyle et al. (2023), who observed increased optimization difficulty as a function of training steps on the RL task. The resulting nonstationarity is strikingly simple: in principle, running a DQN agent amounts to regressing on one-hot image labels and periodically adding a constant offset to the regression targets. We show in Figure 3 that in fact the optimization difficulties encountered by this agent can largely be attributed not to the fact that the target is *nonstationary* but specifically the fact that this nonstationarity *increases the target mean*. Our experiment setting is simple: we construct a fixed learning problem consisting of random network outputs offset by some fixed constant, using as input images from the MNIST dataset. We use this learning problem as a "pretraining" task, and then "fine-tune" the network on a new set of random network outputs offset by either the same constant or zero. We observe in Figure 3 that pretraining on large offsets significantly reduces the network's ability to learn on new targets, whether these new targets have the same mean as the pretraining task or are centered at zero, and that the degree of interference with new tasks increases as a function of the pretraining offset. While layer normalization can mitigate this trend, it does not completely eliminate it, particularly in the case of centered finetuning targets (c.f. Appendix D.1).

## 4    MITIGATION STRATEGIES

Having identified a set of mechanisms which act independently to induce plasticity loss, we now investigate whether mitigation strategies to these mechanisms can be studied independently and then combined to produce more robust learning algorithms. Such an approach bears the promise to significantly reduce the combinatorial complexity involved in the search space of possible interventions.

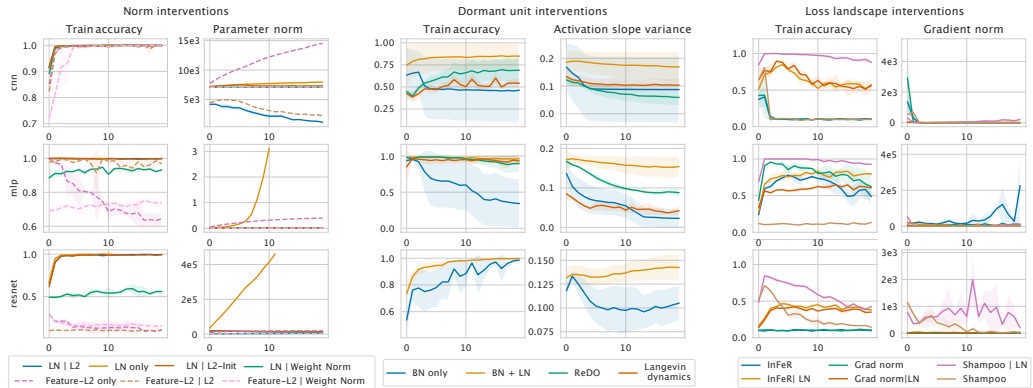

**Figure 4:** Comparison of interventions aimed at addressing different failure modes: overall, combining layer normalization with L2 regularization addressed plasticity loss in all classification problems we considered. Many other strategies also improve over doing nothing, but do not outperform this baseline.

## 4.1 ADDRESSING INDIVIDUAL MECHANISMS

**Unbounded norm growth.** To prevent feature and parameter norms from growing indefinitely, one can either enforce hard normalization constraints or use softer regularization strategies. We evaluate a variety of these in the leftmost subplot in Figure 4. We find that normalizing features along either the layer or batch dimension does not adversely affect network expressivity or learning speed; however, constraining the norm of the weights (in our case, by rescaling the per-layer vectorized weight norm to its value at initialization, a heuristic proxy to path normalization (Neyshabur et al., 2016)) did interfere with optimization speed. Conversely, regularizing the norm of the features was less effective than applying the hard constraint of layer or batch normalization.

**Maintaining unit expressivity.** One natural solution to the problem of dead units is to simply not use saturating activation functions (Abbas et al., 2023); however, this strategy does not resolve the signal propagation issues arising due to preactivation distribution shift. We consider two primary classes of interventions: constraining the pre-activations to be centered at zero along either the batch or layer dimension, and resetting units which have saturated using ReDO (Sokar et al., 2023). We observe in the central subplot of Figure 4 that re-centering has beneficial effects on both plasticity in later task changes and on convergence rates in each task. In contrast, methods which reset dead or colinear features are more robust to plasticity loss than standard training approaches, but sometimes slow down convergence on single tasks and are unable to resolve signal propagation issues in larger networks that arise at initialization, which is why they are not included in our evaluations on ResNets. We include a more detailed sweep over more creative normalization layer combinations in Figure 9 of Appendix D, finding slight improvements over the naive combination considered here.

**Loss landscape conditioning.** The regularization and normalization strategies studied previously occurred on the level of maintaining the expressivity of individual units; however, it is intuitively plausible that interventions on higher levels of abstraction, such as the geometry of features and curvature of the loss landscape, may provide additional benefits. We test this hypothesis in the rightmost subplot in Figure 4, where we consider a squared gradient norm penalty (Barrett & Dherin, 2020; Smith et al., 2021), regularization of a feature subspace towards its initial value (InFeR, Lyle et al., 2021), and the curvature-aware optimizer Shampoo (Gupta et al., 2018). All results are run in networks with layer normalization but without L2 regularization. While we see some performance improvements at the end of training on some nonstationary tasks, none of these approaches consistently outperforms the combination of L2 regularization and layer normalization studied earlier.

## 4.2 COMBINING IMPROVEMENTS

**Supervised learning.** Consistently throughout our preceding evaluations, we have observed significant benefits from the simple combination of layer normalization on all preactivation vectors and L2 regularization on the network's weights. We now expand our evaluations to incorporate input distribution shifts, gradually expanding datasets, and composite forms of nonstationarity where only

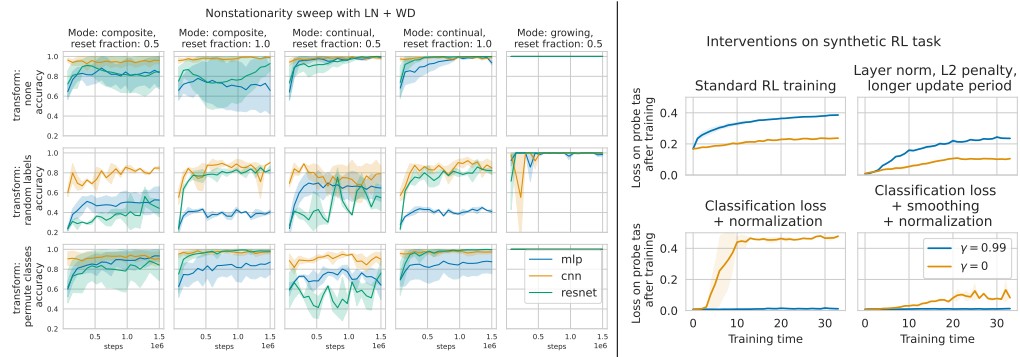

**Figure 5:** Left: Layer normalization and L2 regularization on synthetic non-stationary supervised classification problems. Right: necessity of scale-invariant output parameterization in a simple RL task.

a subset of inputs change over time. We evaluate three architectures: a MLP, a CNN, and a ResNet-18. Each architecture has layer normalization applied before each nonlinearity, and is trained with a fixed L2 penalty of $10^{-5}$ (additional tuning of the L2 penalty can yield better per-task performance, as can be seen in Figure 18). We use the CIFAR-10 benchmark as our base dataset. Each network is trained for 1.5M optimizer steps, which corresponds to 20 tasks. In the 'continual' mode, a random transformation is applied uniformly at random to a fixed fraction of the dataset, either 0.5 or 1.0. In the 'composite' mode, the random transformation is always applied to the same data points. In the 'growing' mode, we start with inputs from a single class, and add inputs from an additional class at each re-randomization, also applying random transformations to the data incorporated so far. Across each of these tasks, we see flat or upward performance trends over the course of training.

**Reinforcement learning.** We take a small digression to revisit a reinforcement learning problem where plasticity loss was previously identified (Lyle et al., 2023). In this environment, the agent is randomly shown an observation from an image classification dataset and a reward of 1 is given if the action taken by the agent corresponds to the label of the image it observes. The resulting optimal value prediction problem amounts to predicting a one-hot vector corresponding to the class label of an image plus a constant bias term (whose value converges to roughly $(1 - \gamma)^{-1}$). We train a DQN agent (c.f. Appendix B.3 for details) on this task, and evaluate the plasticity of parameter checkpoints by regressing from those parameters to a fixed-norm perturbation of the network's current outputs. As observed in Section 3.4, the large target norms induced by the high discount factor result in plasticity loss even in the network incorporating layer normalization.

Distributional losses (Imani & White, 2018; Bellemare et al., 2017; Schrittwieser et al., 2020; Stewart et al., 2023) have been demonstrated to improve performance on a variety of regression problems, and have been found to help mitigate plasticity loss (Lyle et al., 2023). Indeed, we see in the bottom left subplot that this appears to completely mitigate the problem for the $\gamma = 0.99$ agent. The failure of $\gamma = 0$ in the case of classification losses illustrates a more subtle form of saturation compared to the dead units discussed previously: for the case of $\gamma = 0$, the network is trained to fit a low-entropy distribution which is best attained by saturating the softmax output transformation: this saturation results in slower learning on later tasks, but can be mediated by label smoothing (c.f. bottom right quadrant) where we interpolate between a random uniform distribution over atoms and the original two-hot encoding of the scalar regression targets. This trick has the effect of scaling the mean of the target distribution by a constant factor, but because it does so uniformly for all inputs it does not interfere with accurate ranking and is easy to correct for if desired.

## 5    NATURAL NON-STATIONARITIES

Finally, we validate the efficacy of our approach on nonstationarities arising in more realistic learning problems. We consider two classes of learning problems: deep reinforcement learning, and natural distribution shifts in image classification.

**Reinforcement learning.** Many prior works have observed that loss of plasticity can present a barrier to improving performance in deep reinforcement learning (Nikishin et al., 2022; Lyle et al.,

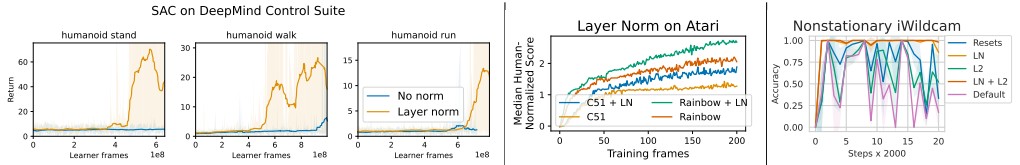

**Figure 6:** Left: performance of a soft actor-critic agent on the humanoid domain from the DeepMind Control Suite using architectures which include or exclude normalization layers. Middle: effect of incorporating layer normalization on value-based RL agents trained on all 57 games of the Arcade Learning Environment benchmark. Right: effect of normalization and L2 regularization on robustness to changes in photo location on the iWildcam dataset.

2021; Dohare et al., 2021; Sokar et al., 2023). However, improving performance in RL requires a delicate balance between maintaining plasticity and not interfering with other aspects of the algorithm. We are therefore interested in evaluating whether the interventions identified in our synthetic experiments are also beneficial in the reinforcement learning setting. To do so, we look at adding normalization layers to a C51 agent trained on the arcade learning environment. This agent combines both the scale-invariant output parameterizations discussed in Section 3.4 and the normalization approach discussed in Section 3.3. We see in Figure 6 that adding layer normalization to Atari agents does consistently provide modest performance improvements over equivalent baselines. Consistent with prior findings (Salimans & Kingma, 2016), we found that batch normalization and L2 regularization interfere with learning; these experiments and per-game performance of all agents can be found in Appendix D. We also validate the beneficial effects of layer normalization on a Soft Actor-Critic agent trained on the DeepMind Control suite. We use fully connected architectures either with or without layer normalization applied prior to each layer as described in Appendix B.2. Here again we find that layer normalization provides significant benefits in several tasks, in particular the `humanoid` domain as seen in Figure 6. Results for the full suite can be found in Appendix D.10.

**Distribution shifts.** We further verify the robustness of our approach to natural distribution shifts using a dataset from the WiLDS benchmark (Koh et al., 2021) which consists of nature camera photos taken from different locations and from different times of day. We train a neural network sequentially on data collected from a single location at a time for two thousand optimizer steps on data sampled from 8 different locations. We randomly sample a new task for 20 task changes. Similarly to the synthetic tasks, we see in Figure 6 that networks trained with L2 regularization and with normalization are better able to deal with changes in the training data distribution, obtaining lower loss on later training sets than networks trained without these interventions. The number of species photographed at a given location, and thus the ensuing difficulty of the task, varies significantly, resulting in learning curves that plateau at markedly different levels and don't present a monotonic rise upwards. Nonetheless, we observe a widening gap between the final loss attained by the networks which use layer normalization and L2 regularization, confirming the efficacy of these interventions. A more detailed visualization which includes an additional baseline and tracks the parameter norm of each network can be found in Figure 20.

## 6 CONCLUSIONS

This paper has shown that while no single network property can explain all instances of plasticity loss in neural networks, a handful of independent mechanisms are responsible for a large fraction of observed cases. Some of these mechanisms are well-known to the community, such as dormant neurons, but others such as large target offsets and preactivation distribution shift had not been previously identified and present exciting directions for future study both as mechanisms of plasticity loss and as potential targets for its remedies. We have further shown that by identifying effective mitigation strategies for each mechanism in isolation and then combining the most effective interventions, we can significantly reduce the combinatorial complexity of the search space of multi-component interventions. We anticipate that this divide-and-conquer strategy may prove useful to future works which aim to find even better mitigation strategies than those identified here. In particular, we note that while L2 regularization is effective at avoiding extreme weight norms which can interfere with learning, it does so at the expense of convergence speed and it is possible that better norm control strategies exist which do not interfere as much with single-task training speed.

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

# A ADDITIONAL BACKGROUND

## A.1 NEURAL NETWORK TRAINING

Neural networks compute a sequence of feature vectors – whose entries are called units – through a sequence of transformations called layers. Common types of layers include affine/linear ones, that typically multiply by a weight matrix and add a bias vector, nonlinear layers, which typically apply an element-wise nonlinear function to their input (called an activation function), such as ReLU or tanh. Also increasingly common, and important to this work, are normalization layers, which normalize their inputs so that either 1) each entry/unit of the vector has sample mean 0 and variance 1 over the current minibatch of training data (called batch normalization (Ioffe & Szegedy, 2015)), or 2) the vector has sample mean 0 and variance 1 across its entries/units (called layer normalization (Ba et al., 2016)).

The weights of the neural networks are initialized to random small values, typically sampled from a zero-mean Gaussian, with two goals in mind: 1) to break symmetries in the network, allowing different hidden units to represent different features, and 2) to allow gradients to preserve norm, typically in expectations, making certain assumptions about the activation function in order to reason about the non-linearity of the system. Training is carried out by randomly sampling initial values of the parameters (typically the weights and biases of the affine layers) and then iteratively optimizing them with respect to an objective function. Objective functions are usually defined as the expected loss over the training set (with often additional regularization terms), and are estimated via iid sampling from the training set for the purposes of stochastic optimization. The most commonly used type of regularization is L2 regularization or Weight Decay (WD), which penalizes the squared norm of the parameter vector. Weight decay can also be incorporated directly into the weight update rule as opposed to being specified in the objective function to encourage the weights to be small in magnitude. The interplay between WD and BN has been extensively studied for a single task setting but remains poorly understood in the context of non-stationary data distribution.

Furthermore, we are specifically interested in studying the effects of the distribution shifts on the network's ability to retain the ability to continually learn. Due to the complexities of the optimization process, various factors contribute to how learning evolves especially in the presence of non-stationarity. In this work, we focus on the following phenomena: 1) *unit saturation* (also referred to as gradient starvation (Pezeshki et al., 2021)) measures the sufficiency of a gradient signal, the lack of which can prevent necessary features from being learned, 2) *pre-activation distribution shift* measures the shift in the pre-activations of a unit which are often initialized to be constrained to zero-mean Gaussian as discussed before, 3) *parameter growth* is aimed at measuring the growth in the magnitude of the norms of the parameters and features as large magnitudes can lead to instabilities in the optimization process, we use the notion of effective dimensionality of a network's representation to quantify this, and finally 4) *loss landscape pathologies* are geared towards quantifying the sensitivity of the network's outputs to changes in its parameters, we use the notion of effective feature rank and its interplay with gradient similarity to measure this formally. We defer the formal details of how these are calculated to the respective sections in the paper later.

# B EXPERIMENT DETAILS

## B.1 NETWORK ARCHITECTURES

We consider three classes of network architecture: a fully-connected multilayer perceptron (MLP), for which we default to a width of 512 and depth of 4 in our evaluations; a convolutional network

with $k$ convolutional layers followed by two fully connected layers, for which we default to depth four, 32 channels, and fully-connected hidden layer width of 256; we also consider a ResNet-18 architecture, which follows the standard architecture.

In all networks, we apply layer normalization before batch normalization if both are used at the same time. By default, we typically use layer normalization rather than batch normalization, although in many ablations we consider various combinations of one or both.

## B.2 CONTINUAL SUPERVISED LEARNING

Our continual supervised learning domain is constructed from a fixed image classification dataset: we have considered CIFAR-100, CIFAR-10, and MNIST and observe consistent results across all base datasets. We primarily use CIFAR-10 in our evaluations. Each continual classification problem is characterized by an input transformation and a label transformation. For input transformations, we use the identity transformation permutation of the image pixels. For label transformations, we permute classes (for example, all images with the label 5 will be re-assigned the label 2), and random label assignment, where each input is uniformly at random assigned a new label independent of its class in the underlying classification dataset.

We fix an iteration interval of 75000 optimizer steps, during which the network is trained using the adam optimizer. This number was selected as it was sufficient for all architectures to converge on most data transformations. We conducted a sweep over learning rates for the different architectures, settling on 1e-5 for the resnet, 1e-4 for the convolutional network, and 1e-5 for the MLP, to ensure that all networks could at least solve the single-task version of each label and target transformation. We then alternate between training the network, and applying a new random transformation to the dataset, for a fixed number of iterations, typically 20.

## B.3 CONTEXTUAL BANDITS

In the contextual bandit tasks, we train a reinforcement learning agent on a stationary environment using Q-learning with target networks as in DQN (Mnih et al., 2015). The environment is based on an image classification dataset (we consider CIFAR-10 in this paper), and is defined as an MDP with observations equal to the images in the dataset, actions equal to the number of classes, and a reward function which yields a value of $\alpha$ if the action taken in the state is equal to its label, and zero otherwise. The state then randomly transitions to a new image from the dataset. While it is not necessary to deploy a reinforcement learning algorithm in order to maximize the reward in this task, it provides a simple setting in which to study the loss of plasticity in reinforcement learning.

The DQN agents we train use target network update periods of either 500 or 5000 steps. We use a replay buffer of size 100,000, and have the agent follow a uniform random policy. In section 3.4 we visualize results for convolutional neural networks, but we also run experiments on ResNets and MLPs. To evaluate plasticity in these agents, we take a network checkpoint, initialize a fresh optimizer, and train the network to fit a perturbation of its initial outputs on a subset of inputs from its replay buffer. Concretely, this means that if we let $\theta_T$ denote the parameter checkpoint we start from, the objective function of the probe task at step $k$ of the fine-tuning phase on some set of inputs $X$ will be equal to $\|f(\theta_T; X) - f(\theta_k; X) + \eta(X)\|^2$, where $\eta$ is a high-frequency function of the input $X$. We run this optimization procedure for a fixed number of steps (2000, selected as this was sufficient for most networks to solve the probe task at a random initialization). This probe task can be thought of as measuring the expected reduction in TD error if the network were to fit its Bellman targets with a random reward function, given some uniform prior over rewards.

**The 'two-hot' trick:** we re-parameterize regression problems as classification problems by first setting some integer bound $M$ on the range of possible values our regression targets might take (for example, in a RL task with $\gamma = 0.9$ and maximum reward 1, we know that the maximum value of a state-action pair will be $M = 10$). We then discretize the interval $[-M, M]$ to obtain a set of atoms $\{-M, -M+1, \ldots, M-1, M\}$. The network's output is a $2M+1$-dimensional vector of logits to a softmax distribution on this set. Given a regression target $c$, we construct a probability distribution with support on $\lfloor c \rfloor$ and $\lceil c \rceil$, with probabilities $P(\lfloor c \rfloor) = \lceil c \rceil - c$, and $P(\lceil c \rceil) = 1 - P(\lfloor c \rfloor)$. We then minimize the cross-entropy loss between this distribution and the softmax distribution output by the network.

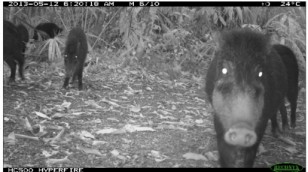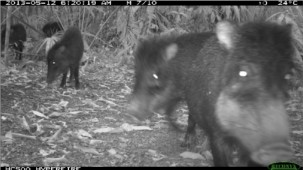

**Figure 7:** Sample from the iwildcam dataset.

### B.4 NATURAL NON-STATIONARITIES

**WiLDS**: The WiLDS benchmark consists of a collection of datasets which can be partitioned to induce distribution shifts. In keeping with prior analyses of image classification, the dataset used in fig. 6 is the `iwildcam` dataset, which consists of photos of animals taken in a variety of locations and times of day. To generate a series of distribution shifts, we train a network only on data collected from a single location; at fixed intervals, we change the location used to generate the training data, and continue training on this new location. We use 10 locations and repeat this process 20 times.

**Arcade learning environment:** we also study deep reinforcement learning agents on the arcade learning environment. While we only consider single-environment tasks, we note that the RL process itself introduces a number of rich forms of nonstationarity, in both the distribution of inputs and the learning targets. We study networks trained with the C51 algorithm, a distributional approach which approximates the probability distribution of returns from a given state-action pair, rather than their expected value, using a categorical distribution on 51 atoms. The learner then minimizes the KL divergence between its predicted distribution and the *distributional Bellman target*. The precise details of this algorithm can be found in Bellemare et al. (2017).

Its relevance for us is that, similarly to the two-hot trick, it leverages the scale-invariant cross-entropy loss in order to solve what is ultimately a regression problem. This allows us to verify that, as in the contextual bandits discussed previously, combining layer normalization and scale-invariant losses can allow the network to maintain its ability to adapt to changes in its prediction targets. We further note that, unlike the two-hot loss, distributional objectives are less likely to struggle with saturated softmax outputs that proved to be a challenge in the contextual bandit tasks because the distributional Bellman targets will typically exhibit nontrivial entropy.

## C ADDITIONAL ANALYSIS

In this section, we include additional ablations and finer-grained analyses relating to the results in the main paper.

### C.1 VISUALIZING LEARNING CURVES

While most of the figures in this paper only show accuracy at the end of a task iteration, we also visualize what the loss curves look like at a more fine-grained level in fig. 8. Note that in many cases, declines in plasticity later in training often correspond to learning curves becoming shallower, rather than to higher plateaus.

### C.2 NORMALIZATION LAYER COMPONENTS

In fig. 9 we ablate the roles of the different components of normalization layers. In particular we are interested in disentangling the relative importance of centering the preactivation distribution about zero, and rescaling to unit standard deviation. We find that most of the performance gain from normalization layers can be attributed to the second mechanism, although the relative effect sizes of each does vary across normalization dimension, network architecture, and dataset.

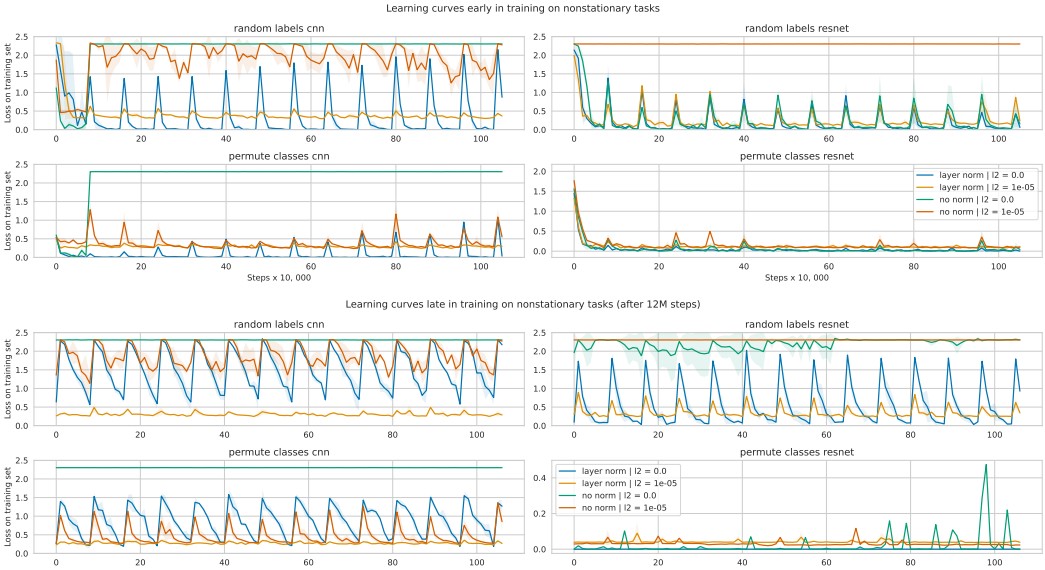

**Figure 8:** Learning curves early and late in a nonstationary training evaluation.

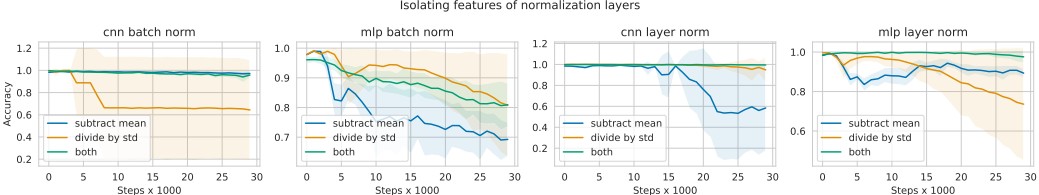

**Figure 9:** Ablating role of mean subtraction and normalization by standard deviation.

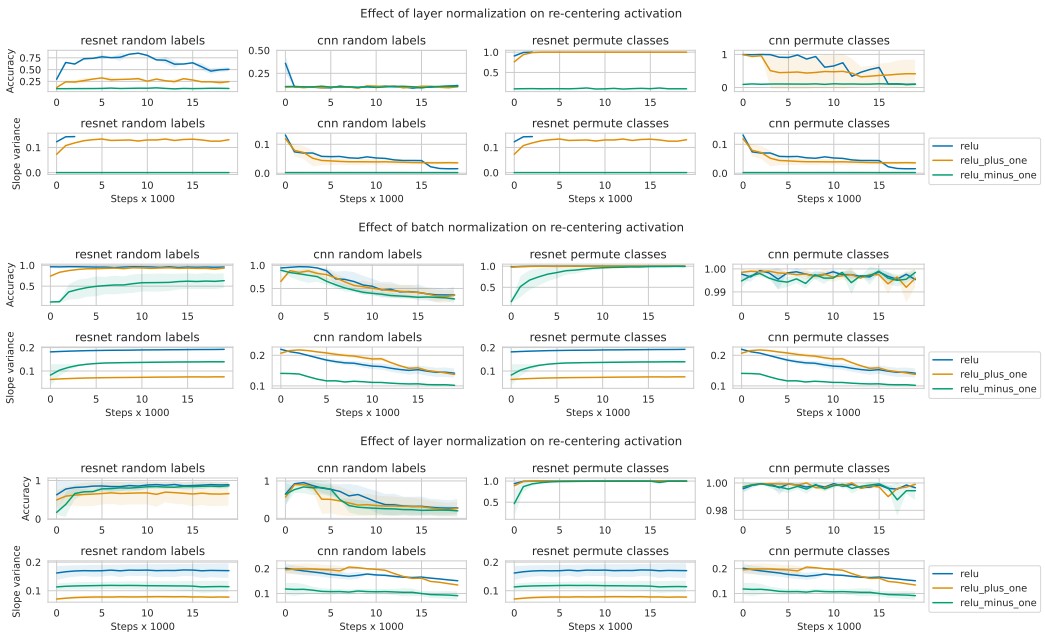

**Figure 10:** Effect of re-centering ReLU units on plasticity and expressivity.

### C.3 INTERACTION BETWEEN NONLINEARITY AND NORMALIZATION

In fig. 10, we evaluate the importance of normalization layers centering their normalization precisely around the region where most nonlinearities exhibit the greatest effect (i.e. zero in the case of ReLU, GeLU, tanh, and leaky ReLU networks). In particular, we see whether adding normalization layers still exhibits a similar benefit to plasticity when the normalization centering occurs away from the 'receptive field' of the nonlinearity. We study this in the context of ReLU units where we offset the input by a fixed value, either $+1$ or $-1$. Intriguingly, while we do observe a reduced variance in the slope of the nonlinearity when an off-center activation is used, this does not always result in reduced performance. Indeed, in many cases the normalized networks outperform an unnormalized network on the tasks despite reduced variability in the behaviour of the nonlinearity. This observation suggests that the role of normalization layers in maintaining plasticity may have less to do with the precise range in which they keep the activations, and more to do with the stability of that range.

### C.4 EFFECT OF DEPTH AND WIDTH

In fig. 11 we investigate the relationship between network size and robustness to nonstationarity. We do not perform any additional hyperparameter sweeps for the different depths. Perhaps surprisingly, we find that in spite of using a learning rate that was tuned for a depth 4 network, using a depth of up to 12 results in improved performance in the permute-classes setting. In contrast, the deeper networks exhibit greater difficulty even in the first iteration of the random labels task. In contrast, and in agreement with the observations of Lyle et al. (2023), we do see consistent improvements from increasing width.

## D ADDITIONAL RESULTS

### D.1 OUTPUT SCALE AND PLASTICITY

We illustrate the effect of increasing $\gamma$ on the plasticity of deep Q-learning agents in Figure 3. While increasing $\gamma$ increases the scale of the TD targets, it also increases the degree of bootstrapping performed by the network, and it is difficult to disentangle these potentially competing mechanisms from the figure alone. To provide additional support for our claim that output scale can damage the plasticity of neural networks independent of bootstrapping, we distill the target phenomenon down

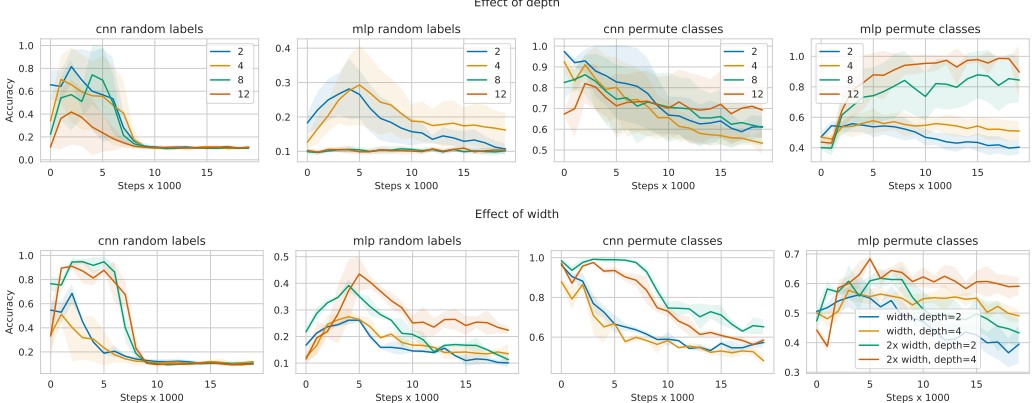

**Figure 11: Top:** relationship between depth and plasticity. We see that networks benefit from depth in the permute-classes task but not in the random-labels task, where increased depth without additional hyperparameter tuning reduces performance and accelerates the loss of plasticity. **Bottom:** relationship between width and plasticity. Across depths, increasing width is beneficial.

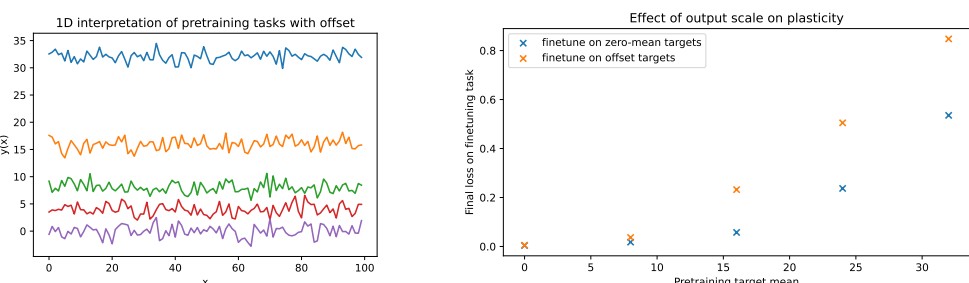

**Figure 12:** Left: Visualization of a 1D analogue the regression targets used for pre-training (actual targets are constructed over an image dataset to allow for more realistic network architectures). Right: "dose-response" curve for pretraining target offset scale vs final performance on fine-tuning task with either zero offset or the same offset as the pretraining data. In both cases, we see similar reductions in fine-tuning performance as a function of the pretraining offset.

to its most essential components. We perform standard mean squared error minimization on a set of random targets to which we add a fixed bias term. In our expeirments, we use the CIFAR-10 dataset as inputs, and construct challenging regression targets by applying a high-frequency sinusoidal transformation to the output of a randomly initialized neural network. Specifically, for $M = 10^5$, we have $f_{\text{target}}(\mathbf{x}) = \sin(M f_{\theta_{\text{rand}}}(\mathbf{x}) + b)$, where $b$ is some fixed bias. A one-dimensional interpretation of the regression targets for different values of $b$ is provided in Figure 12.

In our experimental framework, we pretrain a different convolutional network (the same CNN architecture as is used in the "standard" RL training regime) on a target function defined as above for each $b \in \{0, 8, 16, 32\}$. We then construct a new set of targets, which use a different random target network initialization and have either bias equal to the pretraining bias or $b = 0$. In both cases, we see that the network pretrained on $b = 0$ is better able to reduce the loss on the finetuning targets. We further see a monotone relationship between the scale of the pretraining offset and the final loss of the network. Averaged learning curves over three random target seeds are shown in Figure 13, and averaged final losses (we average both over seeds and over the final 10 SGD iterates to reduce noise) are shown in Figure 12. We run the experiment on a subset of 10,000 images from the underlying dataset and use a batch size of 512; we use the adam optimizer with learning rate 0.001.

Looking into the singular value decomposition of the penultimate-layer features of the network, obtained by performing SVD on a $n \times d$ matrix given by feature embeddings for $n$ randomly sampled inputs with $n = 128$ in this case, gives some insight into what is driving the optimization difficulties in networks trained to minimize a regression loss on a target of the form $f(x) = 100 + \epsilon(x)$. In

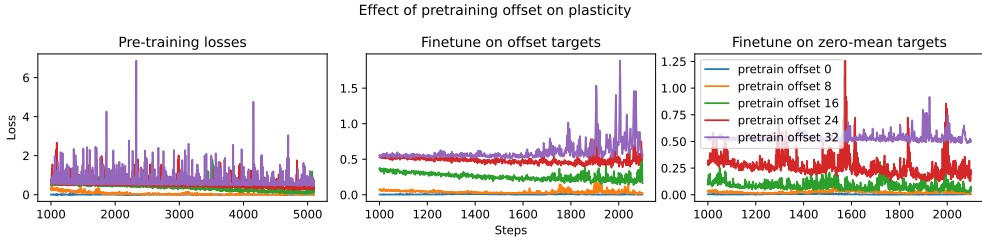

**Figure 13:** Effect of pretraining bias magnitude on finetuning accuracy. We omit the first 1000 steps of the fine-tuning period to allow for a more informative scale.

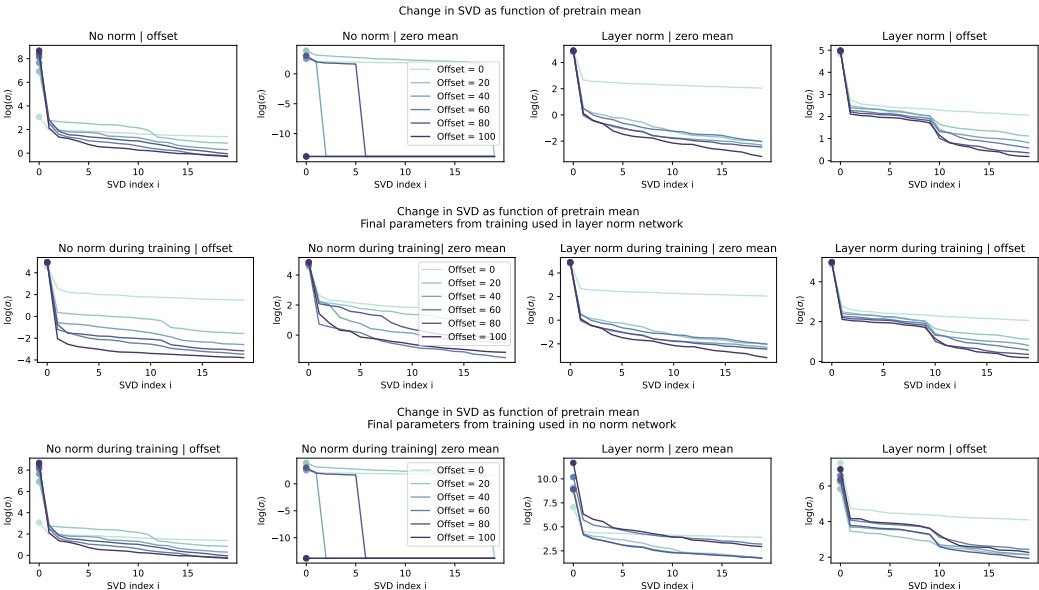

**Figure 14:** Distribution of top singular values as a function of index (note log scale). Magnitude of the top singular value grows with pretraining offset norm and remains high even after fine-tuning on We observe that networks with layer normalization have less extreme ill-conditioning of their features than networks without layer normalization.

networks which don't include layer normalization, we see an immense increase in the maximal singular value as the pretraining target offset grows (ranging from $O(10^3)$ for mean-zero targets to $O(10^8)$ for mean-100 targets). Layer normalization constrains the maximum norm of the features in networks which incorporate it and thus the maximum singular value of the feature matrix, however even in these networks we see a significant decline in the magnitudes of lower-order singular values relative to the largest one.

And what exactly does this largest singular value correspond to in feature space? As we can see in Figure 15, this is the dimension in which the network is encoding the bias term of the targets. First, we note that in theory a neural network with nonzero bias weights in its final output layer should in theory be able to represent functions with arbitrarily large means by increasing the corresponding bias term; however, this does not happen in practice. We observe in the rightmost plot of Figure 15 that the norm of the bias weights in these networks does not exhibit a monotone trend as the pretraining target offset grows. This is particularly striking in contrast to the trends observed for the feature embeddings, which grew not only monotonically but by several orders of magnitude.

Though their bias terms are small, these networks are nonetheless outputting functions whose mean corresponds to that of the pretraining targets, which means that a large constant function must emerge from somewhere. As it turns out, rather than encoding this mean in the bias weights, the networks have instead learned to produce features which can be used as effective bias terms. To

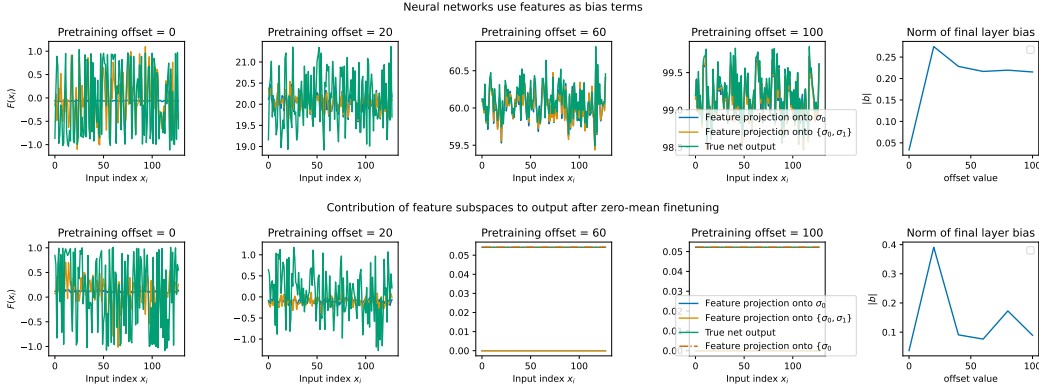

**Figure 15:** Visualization of contribution of principal components of penultimate layer features to the final network output, in contrast with the contribution of the final-layer bias term.

study the contribution of a feature dimension to the network's output, we project the feature matrix onto that vector and then apply the final layer weights to the projected matrix, visualizing the resulting projected outputs for 128 randomly sampled inputs as a (highly disordered) line plot. Strikingly, these visualizations make clear that the subspace encoded by the principal singular value is being used by these networks as essentially a bias term, to which perturbations arising from other subspaces are added. As target magnitude increases, so does the deviation from a perfectly constant function along with the relative contribution of lower singular vectors. Notably, in the networks which are fine-tuned on targets with mean zero, the corresponding dimension of feature space contributes almost zero to the final fine-tuned output, suggesting that the network is using this dimension as an offset term and overwrites it when this offset is no longer needed (though in the case of larger magnitudes this is accidentally achieved by saturating units, as we can see in the second row of Figure 15 that the network output is characterized entirely by the bias).

As a result, we see that enormous singular values and ill-conditioning of the feature embeddings is a direct consequence of outputting a large, near-constant value. This occurs in its most extreme form in networks trained without layer normalization: in networks with layer normalization, this ill-conditioning effect is somewhat mitigated but nonetheless still occurs. Although in theory a network could increase the magnitude of the bias term in its output layer to achieve a larger output mean, in practice networks encode a 'bias' term in their features instead.

### D.2 DEATH OF A RELU NEURON

Figure 2 illustrates at a high level how early gradients are biased towards either reducing or increasing pre-activation magnitudes over all inputs. It is natural to then conclude that, because updates immediately after a task change can be extremely large due to out-of-date second-order estimates in adaptive optimizers such as Adam, this bias will have the effect of potentially killing off many units. In this section, we provide a step-by-step illustration of how units die off, and why sudden task changes are particularly hazardous for ReLU activations. We rely on the visualizations in Figure 16. Concretely, the following four factors drive the death of ReLU units:

1. Immediately after a task change, the network aims to increase the predictive entropy of its outputs. This can be seen in the increase in output entropy immediately following the task change, along with the corresponding decrease in logit norm over the first hundred optimizer steps.

2. There are two ways to reduce the magnitude of the incorrect class probability: one can either reduce the norm of the final-layer weight associated with that logit, or one can reduce the norm of the incoming features. As we can see, the network tends to decrease the norm of the incoming features.

3. Gradients which reduce the norm of the features have the effect of reducing the pre-activation value for most inputs, since negative pre-activations do not contribute to the

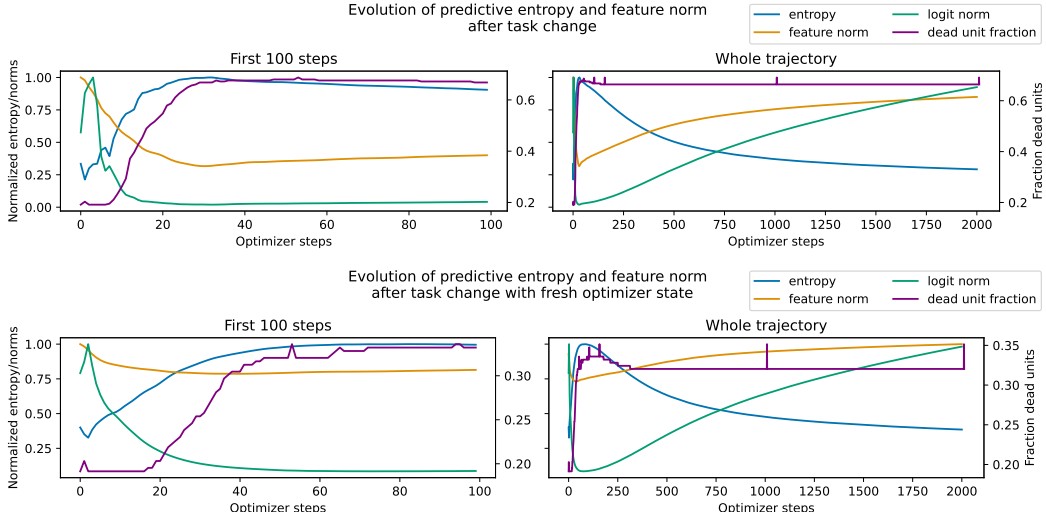

**Figure 16:** Evolution of logit norm, feature norm, logit norm, and dead unit fraction in a one-hidden-layer MLP trained to memorize one set of random MNIST labels, and then fine-tuned on a second set. Results show early and total training dynamics on the second set of random labels. Top: initial optimizer state and parameters taken from the final step of the previous task. Bottom: resets the adam optimizer before continuing training on the second set of random labels. Note the smaller right-hand-side y-axis scale in the bottom figure, corresponding to half the number of dead units.

        feature norm due to the ReLU. We see that in many units the gradient has a negative dot product with all inputs.

4. If the optimizer state is not reset, then outdated second-order estimates cause large update steps. Large update steps in a direction which push down pre-activations quickly results in unit death, with all pre-activations becoming negative within a handful of steps.

### D.3 DETAILED ATARI RESULTS

We provide per-game performance curves for the deep RL agents trained on the Atari benchmark in Figure 24, and also illustrate the nuances of applying weight regularization in this domain in Figure 17. As can be seen from these figures, layer normalization provides obvious benefits in the C51 agent, however the agent does not benefit from L2 regularization. Note that unlike the figure in the main body of the paper, our sweep over L2 regularization coefficients is only evaluated on a subset of the atari benchmark obtained by sorting the games in alphabetical order and selecting every fifth one. Our analysis of the seaquest environment shows that the parameter norm of these agents grows at a modest rate and is several orders of magnitude below the level which caused optimization challenges in the image classification tasks. We see that a penalty of 1e-7 slightly slows the rate of parameter growth without interfering with performance, a value that is much lower than what we studied in the sequential supervised learning setting. This is consistent with prior observations that deep RL agents tend not to benefit from L2 regularization.

### D.4 L2 REGULARIZATION SWEEP

Noting a small gap in peak performance between the models trained with layer normalization but either with or without L2 regularization, we investigate whether this gap might be caused by an overly-aggressive L2 penalty. Given the highly noisy nature of the learning targets in the case of random label memorization, we would expect that maintaining small weight norm would be in tension with reducing loss on the training set, and so may require a smaller penalty than the value of $10^{-5}$ that we used as an initial 'reasonable guess'. We run this experiment on four architectures (we omit the ResNet due to the longer training time required and the short rebuttal period): a depth-4 CNN with fully-connected layer width 256 and either GeLU or ReLU activations, along with a depth-4 MLP with width 512, again using either GeLU or ReLU. We run each network configuration

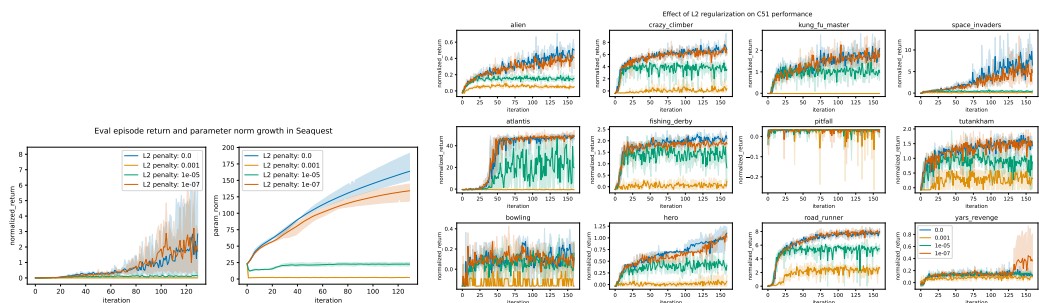

**Figure 17:** Parameter growth in Seaquest and effect of L2 regularization on the C51 agent in a subset of Atari.

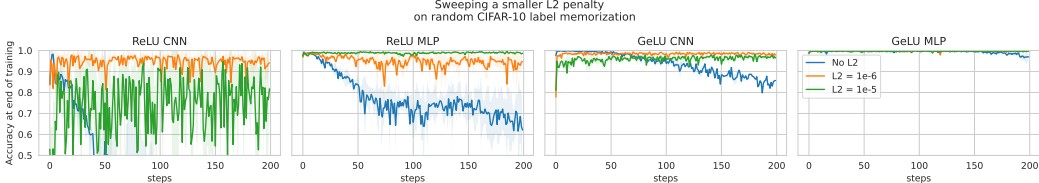

**Figure 18:** Comparison with a smaller L2 penalty: for suitable regularizer weights, we see minimal reduction in performance at the end of an iteration; using a larger-than-optimal regularizer can slow down training in some instances.

for 200 iterations of 200K steps, and use five different random seeds. Shaded region indicates standard deviation. As we see in Figure 18, reducing the L2 penalty by an order of magnitude and allowing for a slightly longer reset interval (200,000 steps) significantly reduces the gap between regularized and unregularized convolutional networks, while for MLPs we see that the 1e-5 penalty is highly effective at maintaining plasticity while maintaining close to 100% final performance even after 200 resets (equating to 40M optimizer steps).

### D.5 LONGER RUN OF CONTEXTUAL BANDIT TASK

Upon request from reviewers, we have included a longer run of the DQN agents trained on the image classification MDP in Figure 19. The agent is trained for 500,000 steps (10x the number of iterations shown in the main body of the paper) in total using the Deep Q-learning algorithm. We sweep over configurations of the following properties: L2 regularization (1e-6 or 0), regression or classification loss, layer normalization or no layer normalization, and either a 'fast' target network update period (500 optimizer steps) or a 'slow' period (5000 optimizer steps). Overall, we see that the networks trained without layer normalization, weight decay, or classification losses quickly saturate at the level of random guessing on the probe task. However, combining these approaches results in significantly better long-term performance on the probe tasks, in some cases resulting in positive forward transfer. There is some variation across configurations whereby networks may see improved or reduced performance on the probe task compared to its value at initialization, but this value tends to stabilize quickly in networks trained in the most stable regimes. In all experiments which use the classification loss, we include a label smoothing parameter of 0.1.

### D.6 WILDS DATASET BASELINE

We also include an additional evaluation of the iwildcam experiment including as a baseline a network which is re-initialized at each task change. Including this baseline, we see that in fact the networks trained with layer normalization not only do not exhibit plasticity loss on this task distribution, but also exhibit positive forward transfer and are able to outperform the reinitialized baseline.

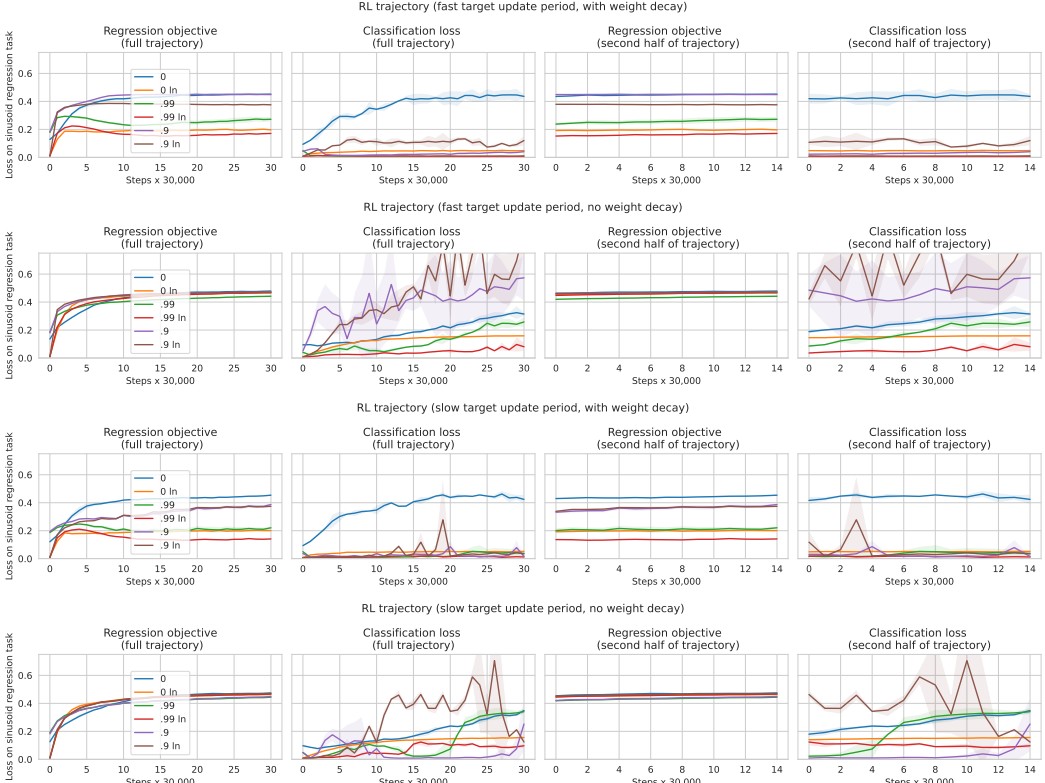

**Figure 19:** Evolution of plasticity under longer runs on the image classification MDP.

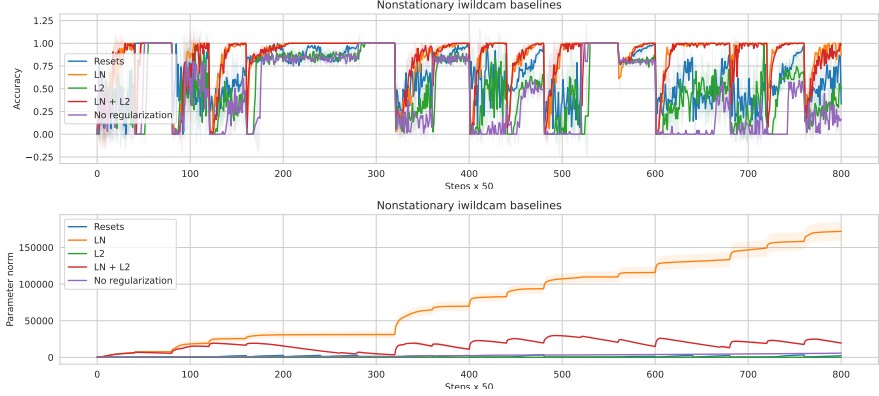

**Figure 20:** We see that using layer normalization allows networks to maintain plasticity and exhibit positive forward transfer on the natural distribution shift problem. The network trained with layer normalization but without L2 regularization exhibits greater parameter norm growth than the other networks, but the training run is sufficiently short that this does not manifest in training difficulties.

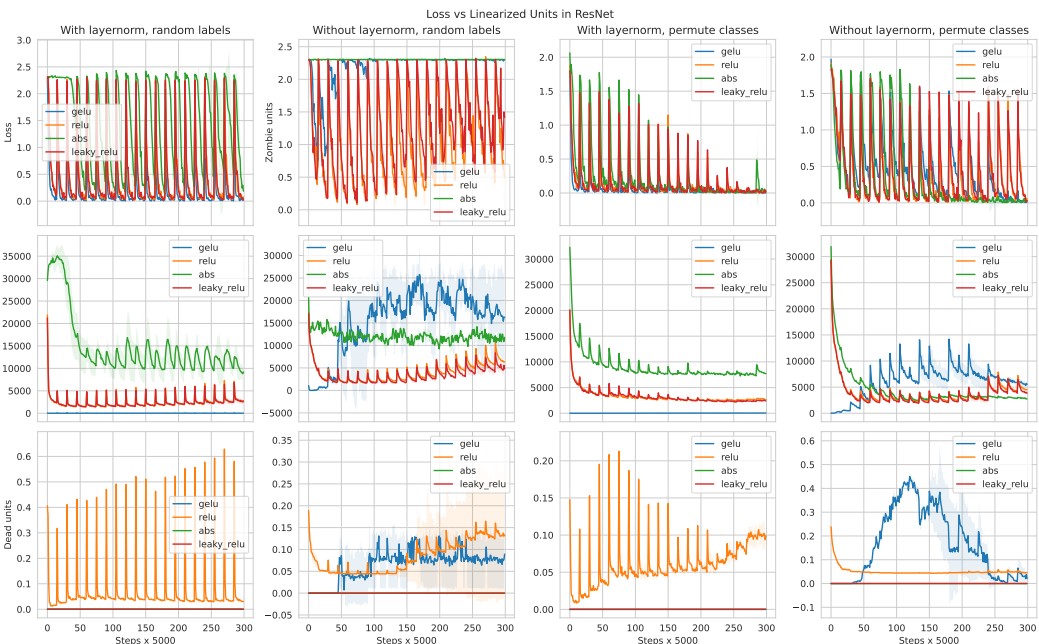

**Figure 21:** Loss of plasticity in our ResNet architecture trained on sequential image classification tasks. We see that high numbers of zombie units tend to correlate with networks that are unable to reduce their loss, even in networks with non-saturating activation functions such as leaky ReLU and absolute value activations. In contrast, dormant or dead neurons are only observable in the ReLU and GeLU networks, providing an incomplete picture of signal propagation failures in the network.

## D.7  UNIT LINEARIZATION IN RESIDUAL NETWORKS

While we provide a hint that linearized or "zombie" units may be more informative than dormant or "dead" units in neural networks in Figure 2, in fact the most compelling evidence that this phenomenon may be tied to plasticity can be observed in residual networks. The ResNet18 architecture we use is significantly deeper than the CNN and MLP, and although it features skip connections which in theory should serve as a buffer against poor signal propagation through blocks, we observe that it tends to be more sensitive to architectural factors such as layer norm than the shallower architectures. This suggests that it is also more sensitive to signal propagation failures, a feature that becomes apparent when we consider a wider range of activation functions. In Figure 21, we consider the ResNet18 architecture under a variety of experimental conditions on the sequential image classification regime. In particular, we consider four activation functions, two configurations of LayerNorm, and two types of nonstationarity.

We see an intriguing correlation between trends in the number of linearized units in the network and the ability of the network to reduce its loss on later tasks: networks with a large number of effectively linear units in some cases are never able to outperform random guessing, even if the network uses a non-saturating activation function and so does not suffer from dormant neurons. Further, within a trajectory, increases in the number of zombie units line up with periods where the network loss also increases, while periods where quantity declines correspond with phases where the network improves its loss (a particularly striking example of this is the case of the absolute value function with random labels and layer normalization). While raw values of the number of linearized units are not necessarily useful for cross-architecture comparisons, it is highly informative within a trajectory, and exhibits significantly stronger correlations with performance even than the number of dead units in the penultimate feature layer.

We also conduct a toy experiment in a small MLP trained on a subset of MNIST with canonical labels in Figure 22. We first randomly initialize the network and train on the subset. We then take the same initialization and map every parameter to its absolute value. As a result, every unit in the second layer of the network is a 'zombie' unit: it is still able to propagate gradients, but only applies

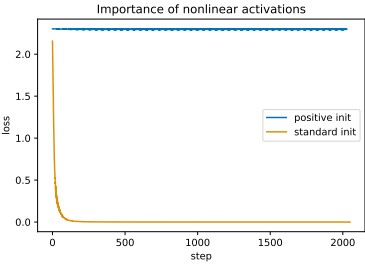

**Figure 22:** Effect of initializing a network to contain only zombie units in its second layer

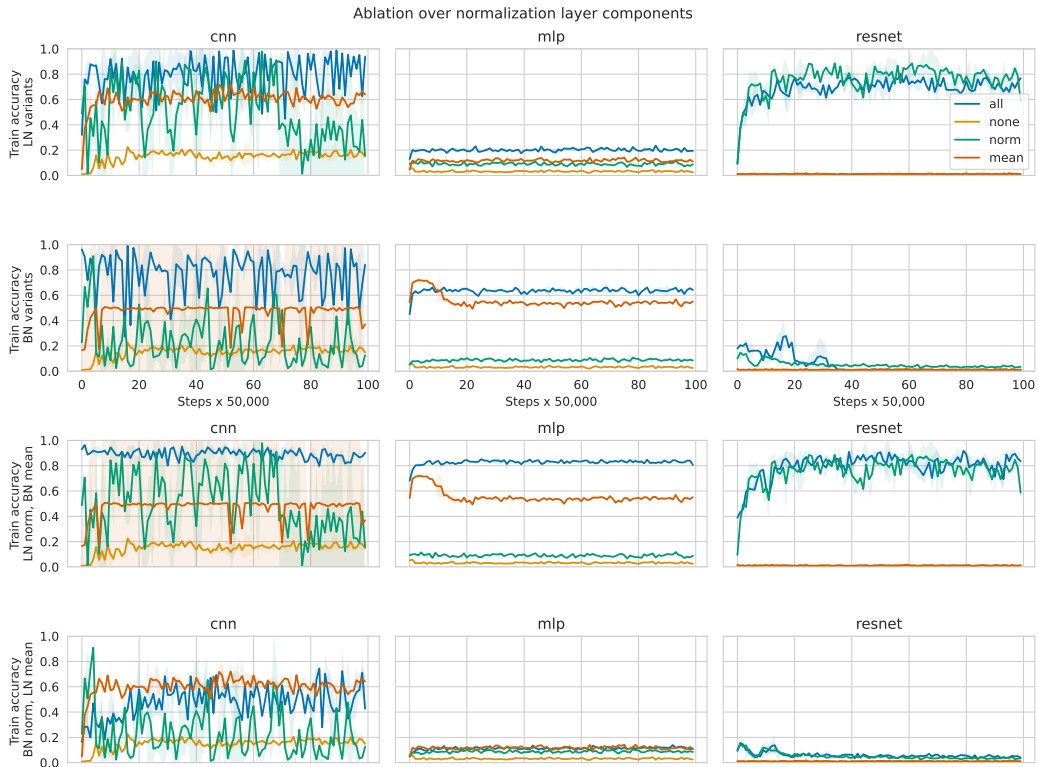

**Figure 23:** Ablations on the axes on which we apply different normalization transformations. We see that applying only the mean subtraction component of batch normalization along with only the standard deviation division of layer normalization results in greater robustness across architectures than applying standard layer or batch normalization.

a linear transformation to its inputs. We find that starting from this initialization, the performance of the network never decreases even after two thousand optimizer steps.

## D.8 Normalization method ablations

Based on the framework outlined in Figure 1, we would expect layer and batch normalization to operate on subtly different mechanisms, and therefore potentially to provide even greater benefits when used in combination. In particular, we would expect that batch normalization would provide particular benefits due to avoiding saturated units and centering the distribution of preactivations to every nonlinearity about zero, which for most activation functions is exactly where the function is most nonlinear, thus avoiding the zombie unit phenomenon. This mechanism is independent of the 'normalization' component of batch norm (i.e. the division by the standard deviation of the preactivations). In contrast, while layer normalization will avoid pathologies where *all* units

saturate, it will have a much weaker effect on this mechanism. We would expect that the benefits of layer normalization would primarily stem from its hard constraint on the feature norm, which will have beneficial effects on signal propagation (Martens et al., 2021) along with a normalizing effect on the feature gradients (Xu et al., 2019). One might then expect that a more effective strategy for maintaining plasticity, at least in supervised image classification tasks which are known to be robust to the stochasticity induced by batch normalization, would be to apply different normalization transformations along different axes of the features.

We explore some variants of this idea in Figure 23, where we follow the nonstationary image classification protocol described previously, using the random label memorization variant. We train each network for 50,000 steps between re-randomizations and use an L2 penalty of 1e-5. We train for 100 iterations. We see architecture-dependent effects, where for example the ResNet is not able to train in the absence of the feature norm constraint from layer normalization while the MLP struggles in the absence of mean subtraction from batch normalization.

In the case of the MLP, we observed in later experiments that centering the inputs at approximately zero was important for training the network, and that most of the benefits of the mean subtraction from batch normalization seen here were replicated by input centering. However, we do consistently observe that across architecture, the combination of division by standard deviation along the feature axis and mean subtraction along the batch axis slightly outperforms naive layer and batch normalization. Ultimately, because the gains we saw from this nonstandard combination were relatively minor in most architectures, and because batch normalization is known to provide much less benefit in RL or natural language tasks than it does in image classification, we opted to use standard layer normalization in most of our experiments.

## D.9    Per-game C51 and Rainbow Agent results

We include a per-game visualization of the C51 agent's performance with and without layer normalization. We include a second variant not included in the main paper which used both batch and layer normalization, noting that our findings corroborate prior observations that batch normalization can hinder performance in deep RL as seen in Figure 24. We also look at the Rainbow agent with and without layer normalization in Figure 25.

## D.10    DeepMind Control Suite Experiment Details

To complement our findings on Atari, we also consider a SAC agent trained on continuous control tasks from the DeepMind Control suite. This benchmark significantly differs from Atari along a number of axes: the action space is continuous, the input space is low-dimensional, and the network architectures typically used in this domain are fully-connected rather than convolutional. Somewhat surprisingly, we nonetheless see significant benefits from the incorporation of layer normalization into the architecture we used. We did not perform extensive hyperparameter tuning on this domain.

**Algorithm:** we use the Soft Actor Critic algorithm (Haarnoja et al., 2018).

**Network:** for the critic architecture, we use a three hidden layer MLP encoder with hidden dimension 256, and put LayerNorm prior to the activations of the first- and third-layer outputs; the output of this encoder is then fed into a two-layer MLP with elu activations. The actor network architecture uses the same encoder architecture with a single 256-dimensional hidden layer prior to the Gaussian policy output head.

**Training protocol:** we train each agent for 1B environment frames and use five seeds for each experiment configuration. We use the adam optimizer with a learning rate of 3e-4 for the actor and 1e-4 for the critic.

**Domains:** we evaluate on 28 domains from the DeepMind control suite.

**Results:** we observe significant gains from layer normalization in several environments where the network constructed without normalization layers fails to make any learning progress.

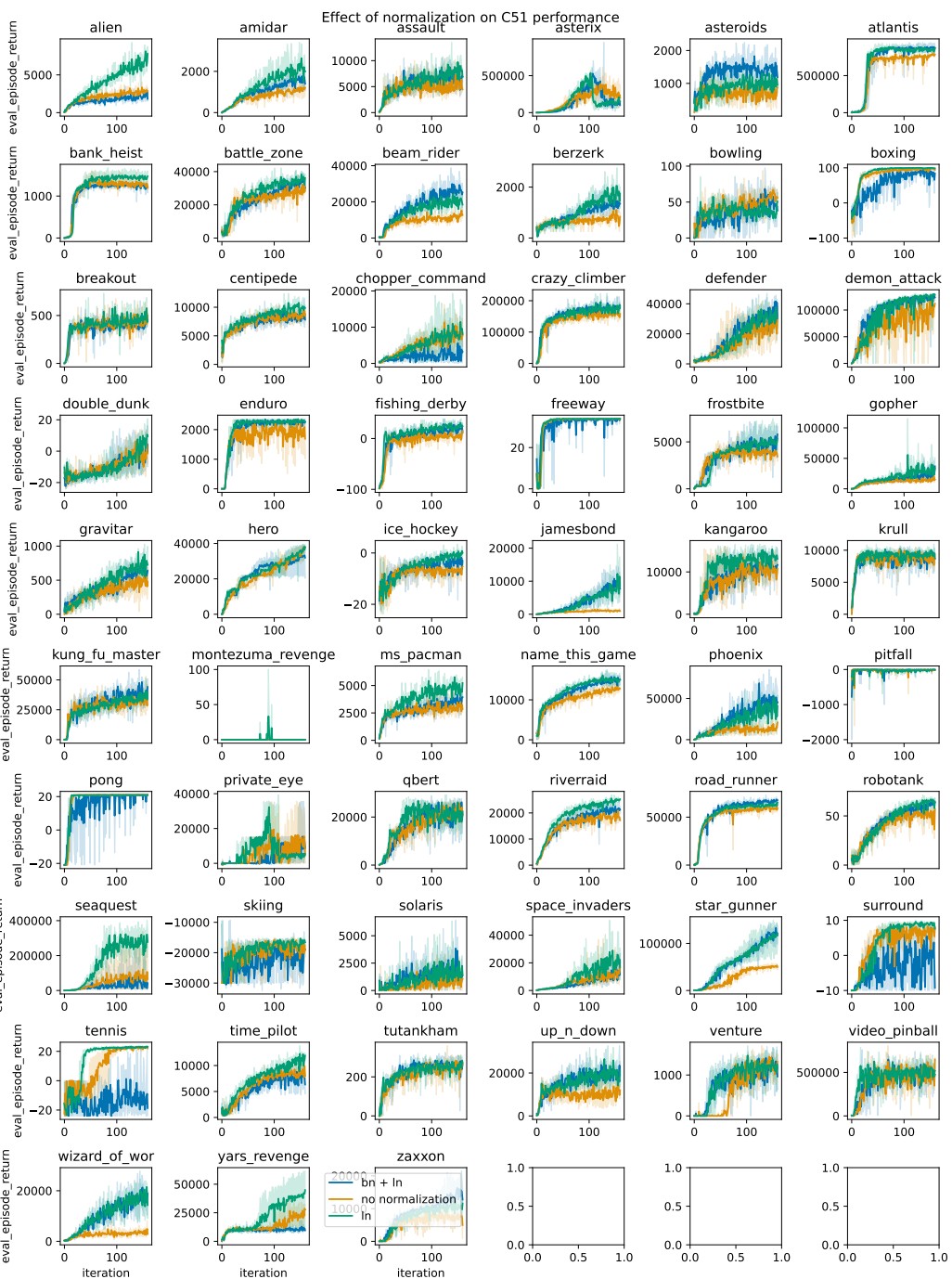

**Figure 24:** Per-game results for C51 agent with layer normalization or a combination of batch and layer normalization.

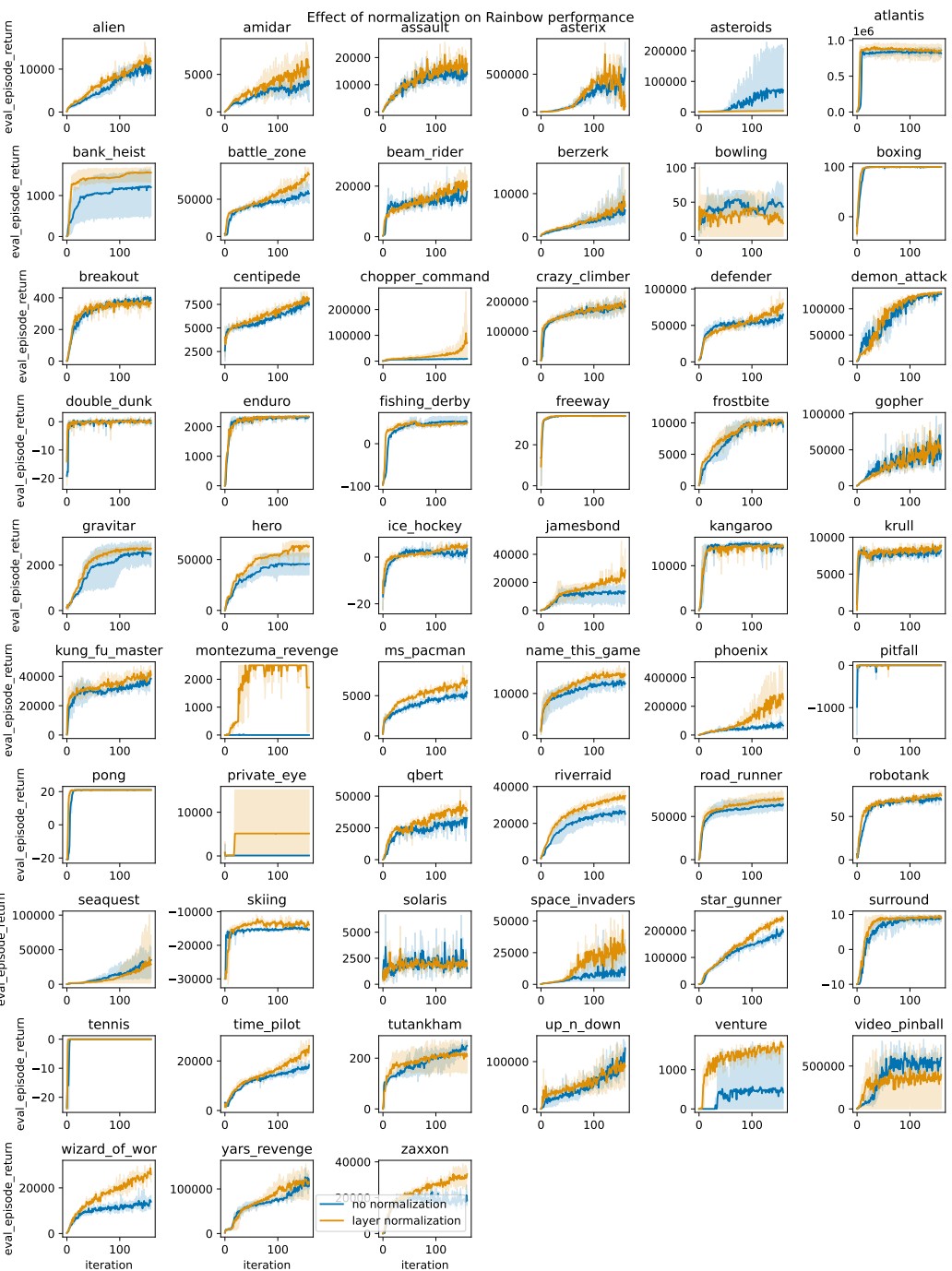

**Figure 25:** Per-game results for C51 agent with layer normalization or a combination of batch and layer normalization.

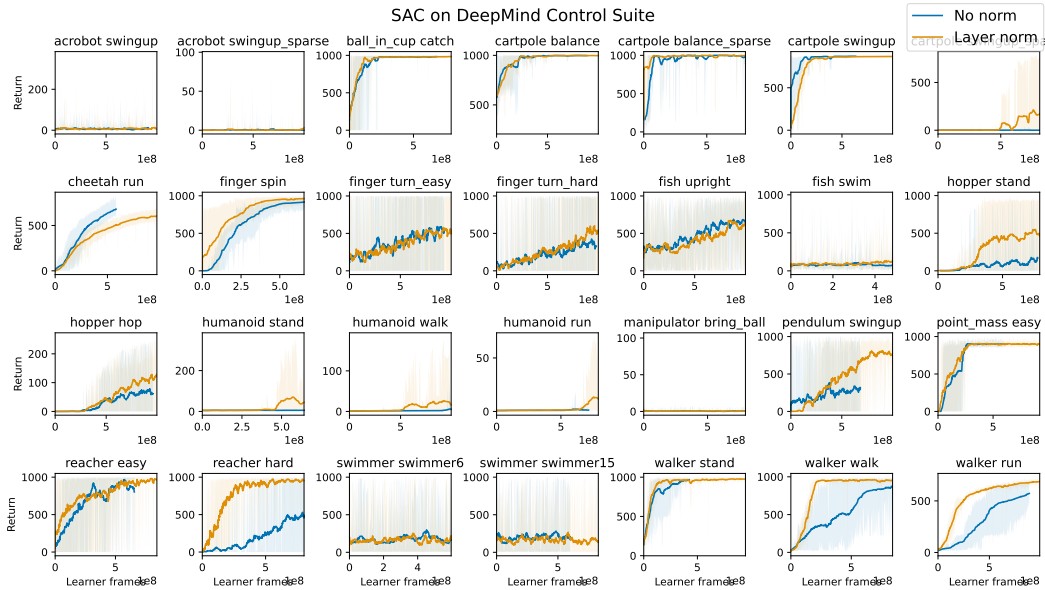

**Figure 26:** Effect of layer normalization in DeepMind Control Suite.

## E A DISCUSSION ON THE INDEPENDENCE OF CAUSAL MECHANISMS

This paper claims that the set of four mechanisms identified in Figure 1 are *independent*. To justify this claim, it is first necessary to clarify what we mean by independence. In this paper we will say that *two mechanisms of plasticity loss are independent if there exist learning problems where completely mitigating one mechanism does not prevent plasticity loss due to the other.* A stronger notion of independence might require that the mechanisms not interact, however in a system as complex as training a deep neural network this is an excessively stringent requirement.

Given this definition of independence, it is then straightforward to determine whether two mechanisms are independent, at least if the answer is positive: it suffices to identify a learning problem constructed so that one mechanism cannot occur, and where the other can be shown to cause the network to lose plasticity. We can in fact find such learning problems within the experiments presented in previous sections, which we now enumerate.

While **preactivation shift** and **weight norm** often correlate in neural network training trajectories, much of this correlation is a natural result of neural network optimization, as both weights and features must necessarily change over the course of learning, and any two quantities which consistently trend away from their initial values may look correlated. To see that the mechanisms by which these phenomena influence plasticity are independent, we first consider a network where the weight norm is regularized via weight decay and can be verified not to grow uncontrollably, but where preactivation shift causes performance degradation. We find such an example in Figure 20, where the network trained with L2 regularization exhibits declining performance, while the network trained with L2 + layer normalization does not. Similarly, the CNN shown in Figure 8 exhibits loss of plasticity in long training trajectories when only layer normalization is applied, but not when layer normalization is used in conjunction with L2 regularization.

In contrast, the relationship between preactivation shift and **unit saturation** is more of a direct causal arrow: it is possible for pathologies to arise as a result of preactivation shift that do not correspond to unit death, for example the linearized units illustrated in Figure 19; however, unit death requires a shift in the distribution of preactivations in order to occur (a counter example would be when units are dormant at initialization, however in this case the problem is not loss of plasticity but rather its nonexistence). Intriguingly, the results of Figure 21 do suggest that, all else being equal, a shift in the distribution of preactivations which preserves the per-unit mean over samples is less damaging than one which preserves the per-sample mean over units, consistent with the unsurprising conclusion

that zeroing out gradient flow through a feature is more damaging than other more subtle signal propagation issues.

**Regression target magnitude** is a unidirectional factor in plasticity loss, as in most cases the magnitude of the regression targets is not caused by other optimization pathologies in the network, but rather is an unavoidable feature of the learning problem. The magnitude of regression targets can exacerbate **preactivation shift**, as large targets require commensurate changes in the network to fit. However, we see that even in networks which feature layer normalization, for which the preactivation mean and variance are fixed, we still see reduced ability to adapt to new tasks as a function of pretraining target magnitude in Figure 13. Similarly, while regression target magnitude can exacerbate **weight norm** growth, only adding L2 regularization is insufficient to mitigate the issue, as we can see in the RL agents in Figure 3 (RHS); similarly, the incomplete success of layer normalization at improving plasticity in this domain suggests that avoiding unit saturation across an entire layer is insufficient to mitigate plasticity loss caused by large target magnitudes.

**Weight norm growth:** it remains to show the independence of parameter norm growth and number of dead units as causes of plasticity loss. This is a relatively straightforward task: weight norm growth is often most pronounced in networks with layer normalization (see e.g. Figure 4), which avoid at least the most extreme version of unit dormancy where an entire layer becomes dormant. In the opposite direction, unit saturation is a highly effective means of reducing the growth of the parameter norm as it prevents gradients from acting on parameters in a way that could increase their magnitude, however has obvious deleterious effects on trainability.

