# OpenReview forum: "Towards Perpetually Trainable Neural Networks"
_ICLR.cc/2024/Conference — Submitted to ICLR 2024_

### Official Review · Reviewer_m9W8 · 2023-10-27

**Soundness:** 2 fair
**Presentation:** 3 good
**Contribution:** 2 fair
**Rating:** 6
**Confidence:** 3

**Summary:**

This paper investigates the problem of plasticity loss in neural networks trained on nonstationary learning problems. The paper conducted a series of experiments and analyses, identifying different mechanisms of plasticity loss, including saturation of nonlinearities, dead neurons, norm growth, and output sensitivity. Consequently, the paper shows that a combination of layer normalization and L2 regularization is a robust mechanism for mitigating all the aforementioned issues. The approach has been validated in an RL setting and on the WiLDS benchmark.

**Strengths:**

- The writing of this paper is very clear and easy to follow.
- The authors conduct a fine-grained empirical analysis of plasticity loss (4 different mechanism, 3 different architectures) and provide detailed explanations on how to interpret them.
-  The authors proposed a training protocol (LN + L2 regularization) for mitigating the plasticity loss.

**Weaknesses:**

The contribution and novelty of the paper are a bit difficult to identify, especially considering previous work such as Lyle et al. 2023 (also cited in this paper), which indicates that factors such as dead units or weight norm may not fully explain the loss of plasticity in learning. Furthermore, normalization and L2 regularization have been studied in the past and are known as ways to prevent plasticity loss (for example, LN was studied in Lyle et al. 2023). Maybe try to extend the current work to a wider RL settings, or focusing on identify newer plasticity loss mechanisms would be better.

Some of the experiments could be questionable or appear to be missing. For example, why is there no L2 regularization used in Figure 6 left (the RL experiments with C51)? Is using a small learning rate equivalent to using a large learning rate? Is using a wider network the same as using a narrower one?

**Questions:**

Please see weaknesses.

---

> ### Author Response · Authors · 2023-11-17
>
> We thank the reviewer for highlighting the breadth of our empirical evaluations and the efficacy of our proposed protocol. We also appreciate the reviewer’s concerns about the positioning of the paper’s contributions relative to prior work and their relative novelty, and hope that our responses sufficiently address these concerns.
>
> **Concern 1.** The contribution and novelty of the paper are a bit difficult to identify, especially considering previous work such as Lyle et al. 2023 (also cited in this paper), which indicates that factors such as dead units or weight norm may not fully explain the loss of plasticity in learning.
> **AR:** We thank the reviewer for highlighting this. In light of this and other reviews, it is clear that we did not sufficiently emphasize the limitations of previous works and how this work overcomes these limitations, and we hope that we have improved the clarity of the contribution in our revisions. We provide a summary here:
>
> 1) We decompose plasticity loss into four relatively independent mechanisms, and show that improvements from mitigation strategies for each mechanism are additive. **The independence of these mechanisms explains the existence of the falsifying examples found by Lyle et al.** For example, mild parameter norm growth does not tend to hurt plasticity, and unit saturation eliminates the potential for parameter growth. As a result, if a population contains some networks with saturated units and others with mild parameter norm growth, one might see a positive correlation between parameter norm and plasticity, whereas in a population without saturating units one would expect a negative correlation. Thus while Lyle et al. (2023) provide a set of negative results showing that no *single* pathology in our framework can fully explain plasticity, this work demonstrates that *combining mitigation strategies for each failure mode can yield significant gains*.
> 2) We evaluate a much **more comprehensive spectrum of mitigation strategies** on a significantly broader set of tasks. We perform a thorough evaluation of varying normalization layer combinations and configurations, and further evaluate a range of strategies to control or mitigate pathological loss landscape curvature;
> 3) We show that in fact the “loss landscape pathologies” identified by Lyle et al. (2023) were largely attributable to the norm of the network outputs (see point 3 below) rather than more subtle properties of the loss landscape. Further, we show that curvature-aware optimization is not sufficient to mitigate plasticity loss, contradicting the implications of Lyle et al.
> 4) Our combined strategies exhibit significant performance improvements over the single interventions studied in previous work, which was unable to completely mitigate plasticity loss in the contextual bandit setting studied in Section 3.4 of this paper, and also shows significant robustness to particularly difficult sequential supervised learning tasks in which which individual mitigation strategies exhibit declining performance. While in light of the framework provided in our paper this combination is straightforward, it is notable that to our knowledge no prior work considered the potential synergistic effects of different mitigation strategies.
>
> A read of the literature prior to this paper would suggest a picture of plasticity loss as a collection of nebulous and ill-defined pathologies of the network which can be partially resolved by disparate intervention strategies, without a clear explanation of what failure modes these interventions are rectifying. This work clearly identifies a set of relatively independent mechanisms of plasticity loss, to which interventions can be combined, and shows that a strategy of studying the effect of each mitigation strategy on its targeted pathology in isolation, then combining the most effective approaches, can yield significant gains over prior methods. We believe that this framework will allow for more effective mitigation strategies to be developed in the future, by more accurately targeting the specific mechanisms of plasticity loss which we have outlined.
>
> **Concern 2** Furthermore, normalization and L2 regularization have been studied in the past and are known as ways to prevent plasticity loss (for example, LN was studied in Lyle et al. 2023).
>
> **AR:** In fact, L2 regularization was also studied by Lyle et al., but it did not give a significant benefit to the RL tasks considered there, and it is not obvious from those results that it would give any additional benefit on top of layer normalization. The observation that weight regularization and layer normalization operate on largely independent mechanisms of plasticity loss is crucial to identify them as a mutually beneficial combination.

---

> ### Author Response · Authors · 2023-11-17
> **Part 2**
>
> **Concern 3** Maybe try to extend the current work to a wider RL settings, or focusing on identify newer plasticity loss mechanisms would be better.
> AR: Re: wider RL settings, given the context of reviewer xE98’s concerns that we evaluated on only a single game, we would like to emphasize that our evaluation included all 57 games from the Atari benchmark. If the reviewer has a particular RL domain that they believe would improve the paper, we would be happy to evaluate on this domain as well. We also emphasize that our evaluation over synthetic nonstationary tasks in Figure 5 is significantly more comprehensive than prior works, including a variety of nonstationarities with varying degrees of smoothness and structure.
>
> Re: newer plasticity loss mechanisms, we emphasize that the following mechanisms and interventions are novel to this paper:
> 1. In contrast to prior work, which studied individual mechanisms of plasticity loss, this work identifies **new emergent phenomena which can arise from their interaction.** For example, the interaction between **normalization layers and parameter magnitude** in the context of plasticity loss had not been previously identified and is novel to this work.
> 2. This is the first paper to identify the **importance of linearized units** in the loss of plasticity in neural networks (c.f. Appendix D.7 for a more detailed discussion of this pathology).
> 3. The **role of regression target magnitude on network plasticity** had not previously been identified. With the addition of the dose-response curves we have provided in Appendix D (Figures 12 and 13), we are the first to demonstrate a direct relationship between the magnitude of the prediction target bias and the ability of the network to adapt to new tasks.
> 4. The label smoothing strategy we use to avoid saturation of the softmax output layer in the RL agents which employ the ‘two-hot’ trick is critical for avoiding plasticity loss in this domain, and had not been identified in prior works, which were unable to eliminate plasticity loss even with this output reparameterization.
>
> **Q1.** Some of the experiments could be questionable or appear to be missing. For example, why is there no L2 regularization used in Figure 6 left (the RL experiments with C51)?
> **AR:** We initially did not include L2 regularization in our experiments due to general consensus in the RL community that this type of regularization tends to slow down learning. We had also observed in prior experiments that weight norms do grow in the atari benchmark, but not to the extent that was necessary to cause performance degradation in the sequential supervised learning tasks. We have included additional results in Appendix D illustrating the effect of layer normalization in a subset of the full Atari benchmark (Figure 15), which confirms that a) weight norm growth is modest compared to the values observed in image classification domains and b) overly aggressive L2 regularization inhibits performance in this benchmark.
>
> **Q2.** Is using a small learning rate equivalent to using a large learning rate? Is using a wider network the same as using a narrower one?
> **AR:** Prior work has noted that wider networks and smaller learning rates tend to be more robust to plasticity loss (Lyle et al., 2023). We conducted a coarse sweep over width and depth for some architectures in the appendix of the original submission (Figure 11). The role of learning rate is also critical: lower learning rates tend to exhibit less plasticity loss, but also slower convergence. Recent related work (Wortsman et al., 2023; “Small-scale proxies for large-scale Transformer training instabilities”)  also suggests that larger learning rates in smaller models may replicate some instabilities seen in larger models in stationary learning problems. It is possible that analogous results can be obtained in the context of nonstationary training regimes, whereby model scale and learning rate can be calibrated to induce similar failure modes in models of varying sizes. We think this is an exciting area for future research, though it lies outside the scope of this paper to study in full.
>
> We thank the reviewer for their time and are happy to discuss any additional or remaining concerns in light of our response.

---

> ### Comment · Reviewer_m9W8 · 2023-11-22
>
> Dear authors,
>
> Thank you very much for your response and the additional information. I acknowledge that I have reviewed the rebuttal. This additional information makes the paper more complete, and hence I have raised my score to reflect that. However, I still think the contribution and novelty are limited compared to prior work.
>
> Best,

---

### Official Review · Reviewer_yXsB · 2023-10-30

**Soundness:** 3 good
**Presentation:** 3 good
**Contribution:** 3 good
**Rating:** 6
**Confidence:** 4

**Summary:**

The paper describes the ways to overcome degradation of training performance of the neural networks due to loss of plasticity.  The authors identify a number of reasons for the loss of training plasticity: preactivation shift, dead and effectively-linear units, norm growth, and shifts in the target function, and propose the simple ways to mitigate these phenomena.  The authors study the proposed interventions, based upon L2 regularisation and layer normalisation in two settings: deep reinforcement learning and classification

**Strengths:**

- It is very important to address the well-documented (as it is well-supported by the references in the paper) phenomenon of the lack of plasticity

- the paper is well motivated and clearly written, however, there are some questions to be resolved on quality and clarity of some particular aspects (see weaknesses below)

- originality: the work builds upon existing well-known methods such as L2 regularisation and layer normalisation, but builds new insight how these remedies could help solve the problem of learning from non-stationary data

- significance: it can help learn from non-linear data streams, both in reinforcement learning and classification scenarios

- the paper discusses in detail the experimental conditions and therefore, addresses reproducibility well

**Weaknesses:**

Cons:
- Clarity aspects: it is important to distinguish between the aspect of nonstationarity and continual learning, see Q1; the relative performance metric needs to be clarified, see Q2

- Quality aspects: see Q3-Q6

- Originality aspects: While the paper researches extensively into reasons of loss and plasticity, the remedies are based on well-known methodologies (L2 regularisation, layer normalisation) and therefore the message is mostly limited to identifying the remedies as opposed to novel methodological contributions towards solving this problem (see Q7)

**Questions:**

Q1. In the introduction, the paper discusses at length the problem of nonstationarity and continual learning at the same time. Although these problems are connected, I think it is important to distinguish between these two. With the problem of nonstationarity alone, one can see the catastrophic forgetting problem entangled with the plasticity loss. The introduction says, for example: “Our empirical evaluations aim to split the philosophical difference between these perspectives. In the following sections, we will evaluate the ability of a learning algorithm to maintain plasticity in a continual supervised learning task, where we fix some image classification dataset (in our case, CIFAR10) and at fixed intervals apply a random transformation to the inputs and labels.” This difference can be illustrated as follows (the example is a toy illustration of a distinction). Imagine that we have trained a model on 180 degrees rotated MNIST symbols. Then we realised that it was wrong and we actually want it to work on non-rotated ones. We decided to continue training from the last checkpoint where we stopped training for 180 degrees rotated MNIST symbols. At the end, we realise that the performance of the model dropped by X per cent due to the problems in Figure 1. In this case, we are happy with catastrophic forgetting (and don’t aim for lifelong learning), but we are not happy with the loss of performance. To address this point, two actions are suggested: (1) disentangling distribution nonstationarity problem in the introduction (2) calculating the baseline of the last task and all tasks performance: in Figure 6, how would the proposed changes affect the performance if we learnt the whole WiLDS dataset in a stationary distribution fashion? How would the proposed changes affect the performance on just the last task?

Q2. In Figure 6, not sure I understand how is the relative performance calculated? Is it calculated for the most recent task, for all tasks, and relative to what?

Q3. Figure 2, top left: should the axes have values? Does it also make sense to report fraction, not the absolute number?

Q4. “Mild L2 regularization (we use a value of 1e-5) is sufficient to resolve this issue in all three architectures.” Why this choice of the hyperparameter?

Q5. *“this analysis will provide a novel explanation of why units die off.”* Although I see the authors point, I’m not sure it’s an entirely correct description of the contribution. Sections 3.1 and 3.2 indeed shed the light on the process of dying out, but they do so through summarising evidence from existing works, i.e. Lin et al, 2016. What is the contribution is that the authors summarise a number of reasons for loss of plasticity and give the pathways towards improving plasticity.  Or am I missing anything?

Q6. “While saturated units have received much attention as a factor in plasticity loss, the accumulation of effectively-linear units has not previously been studied in the context of continual learning.” Not sure it fully covers such accumulation in the context of continual learning, as I mentioned in Q1 as the task itself does involve learning in face of nonstationarity but not tackling the question whether and how the previously accumulated knowledge should be used (or discarded). Would it be possible to contrast the added aspects of continual learning to previous works, such as Montufar (2013), in this analysis?

Q7: Does different types of layer normalisation (batch normalisation, layer normalisation provide different impact?) Did the authors compare between those different remedies?

---

> ### Author Response · Authors · 2023-11-17
>
> We thank the reviewer for their detailed comments, and for highlighting the importance of the paper topic and insight provided by our analysis. We address specific concerns below:
>
> **Concern 1.** While the paper researches extensively into reasons of loss and plasticity, the remedies are based on well-known methodologies (L2 regularisation, layer normalisation) and therefore the message is mostly limited to identifying the remedies as opposed to novel methodological contributions towards solving this problem (see Q7)
>
> **AR:** We agree that the components of the most effective strategy we identify are already widely known. However, we disagree that this reduces the significance of the contribution: first, as noted in other responses, we emphasize that the primary contribution of this work is the observation that plasticity loss can be decomposed into largely-independent mechanisms, and that mitigation strategies on these mechanisms can be combined. The observation that mechanisms of plasticity loss can be studied in isolation and then combined together significantly reduces the combinatorial complexity of the search for even better plasticity-preserving training algorithms. Second, we note that this unoriginal combination outperforms a number of more “novel” methods which we evaluated and which have been previously published. Showing that a simple existing method can be adapted to outperform newer more complex approaches is in our opinion an important regularizer in the complexity of machine learning algorithms; indeed, we tried a variety of more "novel" approaches (e.g. the normalization strategy in Appendix D, Figure 21) which did not significantly outperform the simple combination of LN + L2 and omitted these from the main paper to simplify the exposition of the work. Finally, the framework we use to select these interventions (the mechanisms presented in Figure 1) can be leveraged to find even more effective solutions: for example, while we use L2 regularization due to its simplicity, it is possible that a more careful variant of weight normalization or regularization towards the initial parameter values may interfere less with convergence speed in single tasks.
>
> **Q1.** Distinguish nonstationarity vs continual learning. With the problem of nonstationarity alone, one can see the catastrophic forgetting problem entangled with the plasticity loss. 2 suggestions: (1) disentangling distribution nonstationarity problem in the introduction (2) calculating the baseline of the last task and all tasks performance in Figure 6, include baseline of training on whole distribution + baseline of training random init each time dataset reshuffled.
>
> **AR:** We use the terms “nonstationary” and “continual” learning interchangeably in this work, and the reviewer raises a valid concern that continual learning problems often evaluate the learner on not just forward transfer, but also on backward transfer to previously seen data (or in its absence, catastrophic forgetting). We agree that this is confusing, and will update the manuscript to ensure that when we evaluate methods on forward-looking performance, we use the term “nonstationary learning”. We have also implemented a baseline for the WiLDS benchmark which resets the network parameters each time the dataset changes, which is included in our revisions (Appendix D, Figure 18).
>
> **Q2.** In Figure 6, how is the relative performance calculated?
> **AR:** We normalize by the human scores used widely in the literature and established by Mnih et al., (2015).
>
> **Q3.** Figure 2, top left: should the axes have values? Does it also make sense to report fraction, not the absolute number?
> **AR:** Having reviewed the figure, the only missing value is the x-axis in the middle subplot, which corresponds to training steps. While switching between fraction and absolute number of dead units does not change the overall trend, we agree that since the total number of units isn’t obvious from the figure, it is clearer to set the y-axis ticks to be a fraction. We will make this change in our future revisions.
>
> **Q4.**  “Mild L2 regularization (we use a value of 1e-5) is sufficient to resolve this issue in all three architectures.” Why this choice of the hyperparameter?
> **AR:**  The value of 1e-5 was chosen because it was the first value we tried and seemed to work reasonably well. In response to this and other reviews, we ran a more detailed sweep over L2 penalties for the MLP and found that in fact a slightly smaller value of 1e-6 results in higher per-task performance while still avoiding plasticity loss in later tasks, this can be seen in Figure 16 in Appendix D.

---

> ### Author Response · Authors · 2023-11-17
> **Response to Reviewer yXsB (Part 2)**
>
> **Q5.** “this analysis will provide a novel explanation of why units die off” but they do so through summarising evidence from existing works[...] Or am I missing anything?
>
> **AR:** We thank the reviewer for highlighting that our initial visualization of the accumulation of dead units was not sufficiently clear. Due to space constraints, we were only able to provide a coarse overview of a nuanced chain of events. In light of this and other reviewers’ comments, we have provided a more detailed visualization of the network trajectory after a task change to illustrate at a step-by-step level how units become saturated. In brief, our explanation consists of 4 observations.
> 1) immediately after a task change, the network aims to *increase the predictive entropy* of its outputs.
> 2) in practice this is done by *reducing the norm of the features*.
> 3) Gradients which reduce the norm of the features have the effect of *reducing the pre-activation value for most inputs*.
> 4) If the optimizer state is not reset, then outdated second-order estimates cause large update steps. *Large update steps in a direction which push down pre-activations can result in unit death*, with all pre-activations becoming negative within a handful of steps. While we attempted to provide a summary of this chain of events in the text of Section 3.1, we agree that it was not sufficiently clear. We illustrate this phenomenon in Appendix D where each step in this chain is illustrated.
>
> **Q6.** “While saturated units have received much attention as a factor in plasticity loss, the accumulation of effectively-linear units has not previously been studied in the context of continual learning.” Issue with saying “continual” rather than “nonstationary” learning. Would it be possible to contrast the added aspects of continual learning to previous works, such as  uMontufar (2013), in this analysis?
> **AR:** Montufar (2013) describes the piecewise linear structure of a neural network assuming ReLU activations, and uses this structure to understand the expressivity of the model. In particular, the focus is on counting the number of linear regions, and providing a specific construction for deep architectures that maximizes the potential number of regions that you get. In this work, **we are interested in learnability rather than expressivity**. We will update the paper to clarify this. In our revisions to Appendix D, we provide a simple demonstration that a completely linearized network can encounter training difficulties (Figures 19 & 20). However, in addition to this example, it is possible to reason more generally about the effect of linearized units on training dynamics with a simple thought experiment, which we include in a child comment.
>
> **Q7:** Do different types of layer normalisation (batch normalisation, layer normalisation provide different impact?) Did the authors compare between those different remedies?
>
> **AR:** We provide one data point on the interaction between layer normalization and batch normalization in Figure 4 of the main paper. However, we also conducted preliminary experiments contrasting the effects of different normalization choices by isolating the effects of mean subtraction and division by standard deviation for both layer and batch normalization, and also studying the interaction between performing different normalization operations on different axes. We did not originally include these in the paper due to space constraints, but we have added a section discussing them and visualizing these results in Figure 21 of Appendix D.

---

> > ### Author Response · Authors · 2023-11-17
> > **Linearized unit example**
> >
> > Let us consider the most simplified version of linearization, where the network finds itself in a state where all units are linear for all data points. Specifically, let us consider a regression on the targets $y = x^2$, with training set $x \in {.5, .8}$ and a model with strictly positive weights (relatively large in norm). In this case, because all relu units are linear the neural network reduces to a linear function parameterized by a product of matrices. The best linear fit to this data would be a matrix with positive numbers (particularly here it is a scalar .68, so if we have a single hidden layer model, the model that obtains this solution would be of the form $\langle w_1, w_2 \rangle = .68$).
> >
> > There are many choices of $ w_1 $ and $w_2$ with only positive values with this property. So taking into account that SGD converges to the local minimum closest to initialization, and assuming the initialization involves only large positive numbers, it is very unlikely for GD dynamics to drive the model to a setting where weight entries are negative (and therefore re-enabling the units to be non-linear), rather than converge to a parametrization of this suboptimal minima where gradients will vanish.
> >
> > One can extend this argument to the level of subnetworks (or individual units). That is, once a unit becomes linear for the entire dataset, and the activation of the unit is sufficiently far away from 0, the parameters for that unit will tend to converge to a local linear minimum. Once the unit has reached such a minimum, there will be no gradient signal to push it into a nonlinear regime. This reduces the capacity of the model to behave in a non-linear fashion. In theory, a distribution shift on the incoming features or gradients might eventually perturb the unit enough to recover nonlinear behaviour, but this is not guaranteed.

---

> ### Comment · Reviewer_yXsB · 2023-11-21
> **Response on the rebuttal**
>
> Dear authors,
>
> First of all, I would like to thank you for a very thorough response to the comments and concerns by multiple reviewers. Sorry for the late response, this is because carefully checking the details of the discussion has taken quite a bit of time.  The revisions of the paper, in my opinion, resulted in a stronger submission. I have checked the responses to myself as well as to other reviewers, and below is my answer to the summary:
> - contribution: the authors stressed that the contribution is not in identifying individual causes of loss of plasticity which have been separately known. In the original review, I mentioned that there is a merit of originality, so I did not actually mean that such empirical analysis of mitigation strategies is not original, it was more a request to clarify the scope, which the authors have done in the revision. As I said in the review, "the work builds upon existing well-known methods such as L2 regularisation and layer normalisation, but builds new insight how these remedies could help solve the problem of learning from non-stationary data". The authors further expanded on the thesis stating that these individual causes are found to be largely independent. More precisely, the authors state that *'the work builds upon existing well-known methods such as L2 regularisation and layer normalisation, but builds new insight how these remedies could help solve the problem of learning from non-stationary data'*. The empirical work, together with a careful empirical support of his thesis, would clarify that it is the independence of mechanisms that is the contribution of the paper. However, see follow-up concern 1 which still needs to be clarified.
> - nonstationary vs continual learning: this concern seem to be explained thoroughly in the rebuttal, which alleviates the original concern.
> - relative performance evaluation: this clarifies on the methodology of evaluation.
> -batch vs layer normalisation: this clarifies upon the original concern.
>
>
> Follow-up concern 1:
> - It does indeed now follow from the paper (see, e.g., Figure 3, Figure 16) that combining the plasticity loss mitigation strategies addresses the issue of plasticity loss; however, is there sufficient evidence to claim that “plasticity loss can be decomposed into largely-independent mechanisms” or should the evidence allow to just state as it is said in the paper: “we find that addressing all mechanisms in conjunction is sufficient to yield negligible loss of plasticity in a variety of synthetic benchmarks.” ? These two claims are different. The second claim looks sufficiently well-supported, while I am not sure about the first one as I am not sure whether it is demonstrated (or how it could be demonstrated that these mechanisms are independent).  I will update the scores if the authors could clarify upon it.

---

> > ### Comment · Reviewer_yXsB · 2023-12-01
> > **Re: revision**
> >
> > Dear Authors,
> >
> > I've just found out that you've actually updated the paper and could not reach out about it and send the answers. Therefore, I will check the revision today and respond and update the scores accordingly.
> >
> > Best wishes,
> > The Reviewer

---

> ### Comment · Reviewer_yXsB · 2023-12-02
> **Explanation**
>
> Dear Authors,
>
> I've read the revision and I am going to update the score accordingly to recommend acceptance as it explains the remaining points well. Saying that, although I see that the authors explained what they mean by the word 'independent', after reading the explanation I agree with the reasoning but I'm still not comfortable with the wording claiming it as 'independent' in the new revision. I don't think it matches the common meaning of this word. However, I don't find it critical, and therefore I didn't take it into account when updating the score.
>
> As I checked, according to the Cambridge dictionary[1], independent means *'not influenced or controlled in any way by other people, events, or things'* which is the stronger sense of the independence according to the paper.
>
> Moving back to the original claim, *'where completely mitigating one mechanism does not prevent plasticity loss due to the other'*, I looked up to the medical literature studying causal effects and they use in similar situation the word 'synergistic' instead [2]. I think if the authors use it as well it would help increase clarity. Cambridge dictionary defines 'synergistic' as *'causing or involving synergy (= the combined power of working together that is greater than the power achieved by working separately): the synergistic effect of two drugs given at the same time'*
>
>
> [1] https://dictionary.cambridge.org/dictionary/english/independent
> [2] https://www.ncbi.nlm.nih.gov/pmc/articles/PMC6662195/
> [3] https://dictionary.cambridge.org/dictionary/english/synergistic

---

### Official Review · Reviewer_xE98 · 2023-10-31

**Soundness:** 2 fair
**Presentation:** 2 fair
**Contribution:** 2 fair
**Rating:** 5
**Confidence:** 3

**Summary:**

Paper presents several failure modes of nonstationary learning, and some solution for mitigating loss of plasticity in neural networks.

**Strengths:**

- Paper studies an interesting and important question - why do neural networks lose plasticity and how to mitigate this problem?
- Includes clear visualizations which summarize experimental findings nicely
- Thoroughly evaluates a variety of different solutions for solving the plasticity problem -- I especially appreciate the breadth of different solutions that were tried in section 4

**Weaknesses:**

Sec 1: Abstract/ Introduction
- Paper claims "[we] identify four primary mechanisms by which neural networks lose plasticity: unit saturation, preactivation distribution shift, unbounded parameter growth, and loss landscape pathologies induced by the network outputs". I believe this is overstating the contributions of the paper, as a number of these mechanisms have previously been identified in prior work.

Sec 2: Related work
-  I think section 2.2 (Loss of Plasticity) needs to be much more specific about the contributions in the prior work, and to more clearly delineate how the contributions of this work differs. In its current form, this section only summarizes how prior work characterize the concept of plasticity in a very abstract manner, without mentioning any specific differences between this work and prior work. However, many of the ideas presented in this paper are very similar to ones already existing in prior work (especially Lyle 2021, Kumar 2023a, Abbas 2023), and it would be helpful for understanding the novel contributions of this paper beyond prior work if it were described in more specific terms

Sec 3: Failure Modes of Nonstationary Learning
- Sec 3.1: while prior works has already identified saturated activations to be a reason for loss of plasticity, this paper claims to provide a "novel explanation of why units die off". However, it is not clear how the experiments in this section supports any novel explanations. More specifically, Fig 2 show that at the beginning of learning a new task, many network representations have negative dot products with all network inputs, while after a few hundred steps, this leads to many dead neurons. This experiment alone demonstrates that saturated activations is a problem for nonstationary learning, but does not provide insights into why. The authors hypothesize (in text) that this behavior may be related to de reduction in confidence on the new task, followed by an increase in confidence after learning the new task, but no evidence was provided for this explanation.
- Sec 3.2: in this section, the paper claims that another reason for the loss of plasticity is the linearization of units. However, there are *no experiments* in this section. Therefore, there is no evidence that this phenomenon happens in practice. Furthermore, the paper itself noted "recovery from this state is possible", so even if it does happen in practice, the network could potentially quickly recover from this state, making it even more important to empirically understand to what extent this "plight" is a problem in practice
- Sec 3.4: this section, the paper hypothesize that another reason for the loss of plasticity is that "regression to targets which have a large mean relative to their variance is innately difficult for neural networks". However, the experiments do not necessarily support this hypothesis. Instead, the experiments show that using distributional losses seem to mitigate loss of plasticity, which is a finding that has already been presented in prior work. While the paper's hypothesis could potentially explain why distributional losses work well, there are a number of other potential explanations, and the paper provide no additional evidence for why the proposed hypothesis is the correct one.

Sec 5: Natural Non-stationarities
- The paper is not specific enough about the exact environments/datasets being evaluated. In particular, the paper says it "trained on the arcade learning environments", as well as "a dataset from the WiLDS benchmark". However, the arcade learning environment includes over 50 games, and the WiLDS benchmark includes 10 different datasets. For the RL experiment, it is unclear whether the experiment is only trained on one environment (if so, which one?), or if the results are averaged over training runs on many different environments (if so, which ones?). Furthermore, the paper should be more specific about which particular dataset it used from the WiLDS benchmark for its experiment.
- Assuming the paper only evaluated on 1 RL environment, I believe this, in addition to only 1 distribution shift dataset, is not extensive enough to validate the efficacy of the proposed approach on realistic learning problems
- The proposed approach is using layer norm with L2 regularization. Why does the RL experiment only compare C51 with layer norm and C51 without layer norm? What about L2 regularization? Even if the L2 regularization doesn't help, or even hurts performance, I think it would
be informative to the reader

Overall, I think this paper has some interesting insights. However, in its current, I believe this paper makes a number of claims which are unsubstantiated by the experimental results.

**Questions:**

See weaknesses section for questions

---

> ### Author Response · Authors · 2023-11-17
> **Response to Reviewer xE98 (Part 1)**
>
> We thank the reviewer for highlighting the importance of the topic area and thoroughness of our evaluations. We also thank the reviewer for highlighting a number of places in the paper where we did not sufficiently convey the novelty of our findings or their positioning to prior work. We are still working on implementing all of these suggested improvements in the exposition of the text, but a preliminary update to the introduction can be seen in the revised pdf. We also thank the reviewer for highlighting points where we did not provide sufficient experimental evidence to be completely convincing – in most cases, the requested experiments had either already been completed and had not been deemed important enough to include in the paper, or were straightforward to run during the rebuttal period, and we have included a significant number of supporting results in Appendix D of the revised pdf. We address the reviewer’s concerns point by point. Note **AR** = author response.
>
>
> **Concern 1:**  "I believe this is overstating the contributions of the paper, as a number of these mechanisms have previously been identified in prior work." **AR:** We agree that some pathologies such as dead units have been identified by previous works. What distinguishes our work is the identification that these mechanisms are largely independent, and that their mitigation strategies can be combined. For example, we disentangle preactivation shift from saturated nonlinearities, whereas prior works viewed these as the same phenomenon. However, avoiding dead units by using a nonsaturating nonlinearity does not mitigate plasticity loss caused by preactivation shift, and previous works which studied unit resets and nonsaturating nonlinearities could not account for this more subtle pathology. We have updated Section 1 to express this more explicitly.
>
> **Concern 2:** Section 2.2 needs to be much more specific about the contributions in the prior work, and to more clearly delineate how the contributions of this work differs.
> **AR:** We thank the reviewer for highlighting this pitfall, and will incorporate into our revisions. As an overview, while our work shares many similarities with that of Lyle et al. (we consider overlapping mitigation strategies and use similar benchmarks), we believe that this paper makes a significant contribution to the literature beyond that provided by prior work. In particular:
>
> a) We show that **mitigation strategies to independent mechanisms of plasticity loss can be combined** to significantly outperform single interventions. Prior work did not consider that interventions may operate on independent mechanisms and benefit from combination. This insight not only explains the falsification results of Lyle et al., who show that single mechanisms fail to explain plasticity loss in all settings, but also provides a more efficient framework for searching for new plasticity preserving optimization methods. The approach we follow, of isolating individual mechanisms, identifying effective interventions on these mechanisms, and then combining the best of each class, significantly reduces the combinatorial complexity of searching over training improvements.
>
> b) Our combined strategies exhibit significant performance improvements over the single interventions studied by Lyle et al. and also shows significant robustness to particularly difficult sequential supervised learning tasks in which which individual mitigation strategies exhibit declining performance.
>
> c) We provide a finer-grained view into the mechanisms of plasticity loss than that provided by previous works, identifying the importance of controlling for preactivation shift and unbounded norm growth. The benefits of layer normalization have been validated in many prior works, but these works **did not provide an explanation of its benefit**. This mechanistic view also sheds light onto the falsification experiments of Lyle et al., giving intuition behind the inefficacy of weight norm in networks with saturated units.
>
> d) we evaluate a much **more comprehensive spectrum of mitigation strategies** on a significantly broader set of tasks. We perform a thorough evaluation of varying normalization layer combinations and configurations, consider recently proposed interventions such as ReDO (Sokar et al., 2023) and Regenerative Regularization (Kumar et al., 2023) and further evaluate a range of strategies to control or mitigate pathological loss landscape curvature.
>
> e) We show that in fact the “loss landscape pathologies” identified by Lyle et al. (2023) were largely attributable to the norm of the network outputs (see point 3 below) rather than more subtle properties of the loss landscape. Further, **we show that curvature-aware optimization is not sufficient to mitigate plasticity loss, contradicting the implications of Lyle et al.**

---

> ### Author Response · Authors · 2023-11-17
> **Response to Reviewer xE98 (Part 2)**
>
> **Concern 3** (Sec 3.2): No experiments on linearization of units. Furthermore…  the network could potentially quickly recover from this state
>
> **AR:** Experiments studying the linearization of units are included in the right hand side of Figure 4, which illustrates how networks trained with non-saturating activations accumulate zombie units in tandem with reduced performance on future tasks, and this accumulation is more pronounced in networks that lose plasticity. While the network could in theory recover from unit linearization, we see in many cases that it does not. Particularly striking is the case of residual networks without layer normalization, where we see that nearly all units are computing linear functions, and for which training curves often never pick up. This can be observed in Figures 19 and 20 in Appendix D. We also note that Appendix C already studies the relationship between plasticity and the variance of the slope of nonlinearities on the preactivation distribution, a quantity which is closely connected to the number of linearized units. We think that further investigation into the effect of these linearized units on optimization dynamics is an exciting direction for future research, potentially drawing on the analysis of Poole et al. and Martens et al. to characterize the feature geometry and gradient dynamics in this network state, and hope that our findings serve as inspiration for further analysis.
>
> **Concern 4 (Sec 3.4):** "regression to targets which have a large mean relative to their variance is innately difficult for neural networks" hypothesis not supported by experiments. Show that using distributional losses seems to mitigate loss of plasticity, already known. No justification for why this is the right explanation, and not just that distributional losses are better.
>
> **AR:** The sweep used in Section 3.4 over the two values of gamma provides evidence supporting our claim that it is the higher target mean, rather than some generic benefit of distributional losses, which drives performance gains. Note in particular that the distributional losses with gamma=0 do *not* improve plasticity over regression losses, suggesting that improvements are not coming from uniform superiority of distributional losses. To illustrate the importance of the target offset value at a finer-grained level of detail, we provide dose-response curves for a MLP trained to regress on targets with varying means and then fine-tuned on a new task in Appendix D, Figure 12. In particular, we see a monotone relationship between the magnitude of the pretraining regression problem offset and the error attained on later tasks.
>
> **Concern 5 (Sec. 5)**: For RL: what exact games are being evaluated? Which dataset from WiLDS benchmark? “Assuming the paper only evaluated on 1 RL environment, I believe this, in addition to only 1 distribution shift dataset, is not extensive enough to validate the efficacy of the proposed approach on realistic learning problems”
>
> **AR:** While the arcade learning environment is only a single benchmark, it contains a rich diversity of games which require complementary algorithmic strengths in order to master; we emphasize that **we evaluate on all 57 games**, and so while we agree that more evaluations always provide more information, it was not clear to us what additional information we would expect to gain from an additional environment. If the reviewer has particular domains outside of Atari that they would like to see evaluated, we would be happy to discuss further. We use the iwildcam dataset from the WiLDS benchmark, as was described in Appendix B.
>
> **Concern 6:** Why no L2 regularization in RL experiments?
>
> **AR:**  Our evaluations on the image classification setting revealed that growth of the parameter norm only begins to interfere with learning when it increases by several orders of magnitude (e.g. from O(10^3) to O(10^7+). While the parameter norm of DQN-style agents on the Atari domain is known to grow, it does not increase by five orders of magnitude (c.f. Appendix C for an illustration of parameter growth in the game Seaquest). L2 regularization is known to slow down training in RL and make it difficult for agents to take off, so we did not anticipate seeing any benefits. For completeness, we provide preliminary results for the C51 agent with L2 regularization on a subset of environments from the Atari benchmark in our updates to the paper (we alphabetized the environments and selected every 5th game), which confirm that L2 regularization does not improve performance in the C51 agent.
>
> We think the narrative around as well as the experimental support for the contributions of the paper have been significantly strengthened by the revisions we have made in response to these comments. If the reviewer has any additional concerns or feels that their original concerns were not sufficiently addressed, we would be eager to discuss further.

---

> > ### Comment · Reviewer_xE98 · 2023-11-21
> > **Response to authors**
> >
> > I appreciate the detailed response from the authors. Many of my questions have been resolved, and I have raised my score. However, I think this paper could benefit from another round of peer review. In the current state, many of the new results are not integrated in the main body of the paper and the related works section has not been updated, and I believe there is much room for improvement in terms of clearly explaining the contributions as well as how the experiments support the claims being made.

---

### Official Review · Reviewer_o35T · 2023-11-01

**Soundness:** 3 good
**Presentation:** 3 good
**Contribution:** 4 excellent
**Rating:** 6
**Confidence:** 5

**Summary:**

This paper provides a good, deep analysis of the loss of plasticity. It provides four mechanisms that lead to loss of plasticity. This leads to the proposal of a training protocol that combines layer normalization, l2 regularization, and scale-invariant output parametrization. The effectiveness of these methods is shown in various supervised continual learning tasks and in some naturally arising non-stationary problems.

**Strengths:**

The paper is well-written and easy to read, for the most part.

The analysis of loss of plasticity is well done and provides various insights into the phenomenon. The proposed methods are well motivated by these insights, and they seem largely effective on a wide variety of problems.

**Weaknesses:**

Although the paper contains good ideas, there are a couple of major issues that stop me from recommending a full acceptance at the moment.

- **Misleading conclusions**
    - The paper's conclusion says, " ... This paper has resolved a critical first step towards robust continual learning algorithms: ensuring that neural networks maintain their ability to learn over time. ...". The algorithms (Layernorm + L2) they propose maintain plasticity but at the expense of peak performance, so it is unfair to say that the proposed algorithm has "resolved a critical first step ."The goal of solving plasticity loss is not to just develop algorithms that maintain a constant level of performance across tasks, but *to develop an algorithm that will maintain the best possible performance for the network.* Consider the extreme case; an algorithm has a learning rate of 0. Clearly, this algorithm maintains plasticity, but it is the worst-performing algorithm. The results in the top-left panel of Figure 3, as well as in Figure 8, show the performance of L2+LayerNorm is worse than the peak of just LayerNorm. L2 is helping maintain plasticity but at the expense of peak performance. I think the conclusion of the paper should be changed to reflect that the proposed algorithms maintain plasticity but sometimes at the expense of peak performance.
    - In the same vein as the point above, the use of classification loss in a regression problem increases the minimum loss that the network can get to because there will be a residual loss for each data point. Even though we might be able to maintain plasticity with a classification loss, it comes at the expense of the best possible performance.
    - At many points, the paper says the previously proposed methods "slow learning in the single-task setting." This suggests that the proposed method has no issues in the single-task setting. But again, that is not true. The proposed method can lead to sub-optimal performance in the single-task case (top-left panel of Figure 3, Figure 8)
- **Missing details of empirical evaluation.** The paper does not provide any details on the number of runs for each experiment, nor does it say what is the shaded region in each plot. It is unclear at the moment if any of the results in the paper have any statistical significance and if they will withstand the test of time.

**Questions:**

- What exactly is scale-invariant output parameterization? This term is used at multiple points in the paper, but it is never explicitly defined. Is it just the conversion to classification loss using the 'two-hot' trick described in section B.3?
- What is "smoothing" on the right side of Figure 3? Similarly, what is "normalization"? Is it just layer norm or something else?
- $\gamma=0$ fails in Figure 3. The paper attempts to describe what is happening there. But I do not understand why the line for $\gamma=0$ loses plasticity.
- Can you please show a zoomed-in image of the blue line ($\gamma=0.99$) of the bottom left plot on the right side of Figure 3? It seems that the loss is increasing for the blue line, but it is unclear from the current image. And if possible, can you run this experiment for ten times longer? If the loss is increasing, a longer experiment will make it clear.
- Why wasn't L2 used in the RL experiments with C51? Is it already included in C51, or did it hurt performance?
- Which figure in the main paper does Figure 8 correspond to?

---

> ### Author Response · Authors · 2023-11-17
> **Response to reviewer o35T (Part 1)**
>
> We thank the reviewer for their thoughtful comments. We appreciate the assessment that our analysis “is well done and provides various insights into the phenomenon”, and that our proposed methods are “well movitated” and “largely effective”. We now address the reviewer’s concerns. Note: AR = Author Response.
>
> **Concern 1:** sub-optimal performance of weight decay.
>
> **AR:**  We agree with the reviewer on the importance of distinguishing between 'not reducing performance' and 'consistently exhibiting good performance', and that L2 regularization can slow down learning in some domains and architectures. We plan to include a more detailed discussion of tradeoffs between performance on a single task and stability in later revisions. The random label memorization task presents a particularly adversarial realization of this tension, as the primary task amounts to memorizing random noise, and L2 regularization tends to have a smoothing effect and makes it more difficult to memorize noisy functions. However, we also note that ultimately this slowdown does not seem to prevent the network from eventually converging on the task -- it just increases the number of optimizer steps necessary to achieve this. To back up this claim, we have re-run a subset of the experiments originally shown in Figure 8 where we provide a slightly longer training interval between label resets, and this longer interval allows both the regularized and unregularized networks to attain >98% accuracy on the task, which can be seen in Figure 16 of Appendix D.  We have also included a sweep over L2 penalty values and activation functions for the CNN and MLP architectures. These results show that the value of 1e-5 which we defaulted to in the paper is suboptimal for certain architecture-dataset combinations, and in some architectures a different value (1e-6 in this case) can recover the stability benefits while also exhibiting minimal performance degradation relative to the network’s unregularized counterpart.
>
> As a result, we do think that L2 regularization at a suitable dose is a valid tool for maintaining plasticity. Our perspective can be summarized as follows: **while L2 regularization can slow down training if applied too liberally, in can be titrated to minimally interfere with performance on the current task while still providing benefits to plasticity by containing otherwise-unbounded parameter growth**. We will ensure that this perspective is effectively conveyed in our future revisions.
>
> **Concern 2:** The use of classification loss in a regression problem increases the minimum loss that the network can get to because there will be a residual loss for each data point. Even though we might be able to maintain plasticity with a classification loss, it comes at the expense of the best possible performance.
>
> **AR:** We are not entirely sure what residual the reviewer is referring to: the lower bound on the cross-entropy loss of the target entropy, or approximation error induced by the binning representation. In both cases, we do not believe that this presents a barrier to the network’s ability to match the desired scalar targets. In the case of the lower bound on the cross entropy, we note that this does not influence optimization dynamics. In the case of expressivity of the binning trick, the distributional output representation does not reduce the expressivity of the network, as any scalar value which a regression model can output, provided it is within the bounded range on which the support of the two-hot distribution is defined, can be represented by a corresponding two-hot distribution (see child comment for details).
>
> **Concern 3:** missing evaluation details.
>
> **AR:** Most tasks run with 3-5 seeds per hyperparameter setting; we will specify these details in our revisions. Shaded regions refer to the standard deviation over the training runs exhibiting the listed hyperparameter values.

---

> > ### Author Response · Authors · 2023-11-17
> > **Details on the two-hot parameterization**
> >
> > We claim that any scalar value $x$ within the pre-selected interval [-M, M] can be accurately represented by a two-hot categorical distribution whose expected value is equal to $x$.
> >
> > To see this, suppose we have defined support on integers in the interval $[-M, M]$, and we have some value $-M < x < M$ which we wish to ‘regress’ the output of the model to. We can write $x$ in the decimal form $k.q$. Then we transform $x$ into the distribution which assigns probability $q$ to atom $k+1$ and probability $1-q$ to atom $k$. This distribution has expected value exactly equal to $x$. Further, while the naive approach which assigns zero probability to most atoms of the distribution can never be fit exactly by a softmax output, the label smoothing trick we use does allow for the network to fit this distributional target exactly.
> >
> > In particular, we obtain the target distribution for which $p(z) = \frac{\alpha}{2M + 1}$ if $z \not \in \{k, k+1\}$ and for the atoms corresponding to $k+1$ and $k$ $p(z)$ is equal to $q(1-\alpha)$ and $(1-q)(1-\alpha)$ respectively. For a given smoothing value alpha, the resulting distribution will thus have expected value $\mathbb{E}[Z] (1-\alpha)$, where Z is the non-smoothed target distribution. It is then easy to correct for this constant scaling factor to obtain the actual scalar value which is being represented. As a result, for every real number $x \in [-M, M]$ we can recover a distribution whose expectation is a fixed scalar multiple of $x$ and which is realizable by a softmax distribution with finite logit norm, i.e. has no residual loss. While in practice finite step size optimization will never converge to this precise distribution, this would not be the case for scalar regression either and so we do not lose any representational capacity by choosing this representation.

---

> > > ### Comment · Reviewer_o35T · 2023-11-23
> > >
> > > Dear Authors, thank you for your thorough response. I appreciate your reply; it has reduced many of my original concerns. I've decided to maintain my recommendation of marginal acceptance as the paper still requires significant rewriting to update the claims about L2 regularization and update the related work section.

---

> ### Author Response · Authors · 2023-11-17
> **Answers to Questions**
>
> **Question 1:** What exactly is scale-invariant output parameterization? Not explicitly defined – just two-hot trick?
>
> **AR:** Yes, the scale-invariant output parameterization we use is the “two-hot trick” in the contextual bandit experiments. The C51 agent, however, uses the standard categorical distributional RL update.
>
> **Question 2:** What is "smoothing" on the right side of Figure 3? Similarly, what is "normalization"? Is it just layer norm or something else?
> **AR:** In figure 3, normalization refers to layer norm, and smoothing refers to the label smoothing approach described in Appx B and in our previous comment.
>
> **Question 3:** Figure 3 concerns about potential uptick. Run for 10x longer?
> **AR:** We have conducted a more expansive sweep over experiment configurations and included the results in Figure 17 in Appendix D. We note that in general, networks trained with layer normalization and weight decay see plateauing loss curves on the probe task which suggest that the network performance has stabilized. These curves plateau at a value comparable to that of the random initialization in most experiment configurations when these three interventions are combined, though there is some variation: some configurations do observe mild plasticity loss and others exhibit positive forward transfer, resulting in slight discrepancies from the loss obtained by a random initialization on the probe tasks. However, in all cases these changes are dwarfed by the plasticity loss incurred by the naive Q-learning algorithm without regularization or normalization layers.
>
> **Question 4:** Why wasn't L2 used in the RL experiments with C51? Is it already included in C51, or did it hurt performance?
> **AR:** Our evaluations on the image classification setting revealed that growth of the parameter norm only begins to interfere with learning when it increases by several orders of magnitude (e.g. from O(10^3) to O(10^7+). While the parameter norm of DQN-style agents on the Atari domain is known to grow, it does not increase by five orders of magnitude (c.f. Appendix D for an illustration of parameter growth in the game Seaquest). L2 regularization is known to slow down training in RL and make it difficult for agents to take off, so we did not anticipate seeing any benefits. For completeness, we provide preliminary results for the C51 agent with L2 regularization on a subset of environments from the Atari benchmark in our updates to the paper in Appendix D (we alphabetized the environments and selected every 5th game), which confirm that L2 does not provide a significant performance benefit.
>
> **Question 5:** Which figure in the main paper does Figure 8 correspond to?
> **AR:** Figure 8 provides a fine-grained view of the learning curves from a subset of nonstationarities from Figure 6; rather than visualizing only the final loss, it shows the per-task learning curves.

---

### Author Response · Authors · 2023-11-17
**Thanks to reviewers, revisions uploaded.**

We would like to thank all reviewers for their extremely helpful comments. Due to the relatively short rebuttal period, we have uploaded a preliminary revision to the paper which contains an updated statement of contributions and an additional section of the appendix (Appendix D) where we have put all additional experiments and visualizations. We are still implementing more subtle updates to the paper to improve the exposition and clarity of the contributions, and will post a second update including these before the end of the rebuttal period. However, to give the reviewers a chance to review the additional supporting results and to voice any additional concerns, we have opted to upload key experimental results a few days ahead of the deadline. If the reviewers have remaining concerns after reviewing the requested changes, we will be happy to discuss further. We think the requested changes have significantly strengthened the paper and look forward to engaging throughout the rest of the rebuttal period.

---

### Meta-Review · Area_Chair_JpZP · 2023-12-10

**Metareview:**

The authors explore the problem of plasticity loss in neural networks, which happens in situations of nonstationarity. Through analyses and experiments the authors obtain insights on the problem which then lead to the identification of techniques for mitigation during training.

This is certainly a very key and important area of research. The paper is motivated well: firstly identifying the problem and then using the insights to suggest a solution. The authors have experimented with different solutions. Overall, the paper's significance is increased by the combination of extracted insights and suggestion for a training protocol.

On the other hand, the reviewers raised some concerns. Many of those have been addressed in the rebuttal period - the authors submitted their rebuttal late but all reviewers checked the rebuttal and replied. However, key concerns have not been addressed satisfactorily in the rebuttal:
- __Claims, conclusions and experiments__: The reviewers believe that even after the rebuttal some of the claims might not be supported by strong arguments. For one, the paper is not placed well in context of rich prior work. Secondly, some of the insights do not align well with the experiments, e.g. not answering the "why". Finally, one reviewer mentions specific issues with the claims, such as using classification loss for regression tasks and potential tradeoff with peak performance; while the authors provide explanations, it seems that the rebuttal and new experiments shed light in some intuition but are not conclusive.
- __Novelty__: This is related to the above, but even post-rebuttal many reviewers believe that the novelty is limited compared to previous work. One of the reviewers who expresses this concern additionally specifies that the paper would have been stronger if the authors used the insights to suggest a novel methodology for maintaining plasticity, rather than identifying existing techniques.

Given the above, it seems that this is an interesting paper but it would need to be updated and go through another round of peer review.

**Justification For Why Not Higher Score:**

Several concerns exist post-rebuttal, the paper needs to be updated.

**Justification For Why Not Lower Score:**

N/A

---

### Decision · Program_Chairs · 2024-01-16

Reject